

# GraphFlow v1.0: approximating groundwater contaminant transport with graph-based methods - an application to fault scenario selection

Léonard Moracchini[1,2], Guillaume Pirot[1,3], Kerry Bardot[4], Mark W. Jessell[1,3], and James L. McCallum[4]

[1]The Centre for Exploration Targeting, School of Earth and Oceans, The University of Western Australia, Perth, Australia
[2]Mines Paris Tech, École Nationale Supérieure des Mines de Paris, France
[3]Mineral Exploration Cooperative Research Centre (MinEx CRC), School of Earth and Oceans, University of Western Australia, Perth, Australia
[4]School of Earth and Oceans, The University of Western Australia, Perth, Australia

**Correspondence:** Guillaume Pirot (guillaume.pirot@uwa.edu.au)

**Abstract.** Groundwater contaminant transport problems remain challenging with respect to their computing requirements. Thus, it often limits the exploration of conceptual uncertainty, that is mainly related to large scale structural features and due to limited characterization. Here, to facilitate geological conceptual uncertainty exploration, we develop further the use of graph representation for geological models to approximate groundwater flow and transport. We consider a faulted multi-

heterogeneous-layer medium to test our approach. The existing rank correlation between shortest path distribution from a contaminant source to the model domain outlet and cumulative mass distribution at the outlet enables to perform scenario selection. The scenario selection approach relies on a metric combining the Jaccard dissimilarity and the Wasserstein distance to compare binary images. Among a set combining eight alternative scenarios, where three faults can either act as a flow barrier or a preferential path, we show that the use of graph-approximations allows to retain or reject scenarios with confidence as

well as to estimate the individual probability of a fault to act as a barrier or a path. This methodology framework opens up possibilities to explore more thoroughly conceptual geological uncertainty for processes affected by flow and transport.

## 1   Introduction

The study of contaminant transport is crucial to understand the fate and behavior of pollutants in subsurface heterogeneous environments. Contamination of groundwater poses significant risks to public health and the environment. It necessitates robust

predictive models to inform mitigation strategies (Burri et al., 2019; Ostad-Ali-Askari and Shayannejad, 2021). Traditional approaches, such as solving partial differential equations (PDEs) for flow and transport, have been extensively used to model groundwater systems (Bear and Cheng, 2010). In particular, MODFLOW is a modeler and solver of differential equations for hydrogeology developed by the US Geological Survey, which is widely used in the research community. However, these methods often require high computational resources (Karmakar et al., 2022).

In recent years, new data-driven approaches have emerged as surrogates for contaminant transport simulation. On one hand, entirely data-based structures have been developed, using various deep learning architectures like transformers (Bai and Tah-



masebi, 2022; Pang et al., 2024). On the other hand, there have been recent attempts involving hybrid models like Physics-Informed Neural Networks (PINNs), which also include differential equations and boundary conditions as inputs (Meray et al., 2024). In both cases, the results are promising, but the number of simulations required for model training and the lack of

transferability remains challenging (Luo et al., 2023). Additionally, tests have mainly been conducted in 1D or 2D due to the significant complexity involved in 3D simulations (Meray et al., 2024).

Graph theory offers a promising alternative to traditional PDE-based models by simplifying the representation of complex systems, without the costly training of data-driven methods. For this approach, the first step is to create a graph to represent a geological model. The choice of vertices, edges, and their weights is crucial. Next, an algorithm, often for shortest path

calculation or maximum flow, is applied to the graph. In recent years, these graph-based methods have been primarily used for studying fracture networks (Hyman et al., 2018; Karra et al., 2018; O'Ghaffari et al., 2011). In such cases, each intersection between fractures is modeled by a node, and geometric and geological information is stored in the edge weights. The use of graphs is particularly effective for Discrete Fracture Networks (DFN) due to their high structural complexity, with the number of elements often being too large to be solved by traditional finite element methods.

Other studies use a graph-based method to approximate the path of minimal hydraulic resistance (or maximal hydraulic conductivity) in a heterogeneous medium. Graphs are generated with hydraulic resistance as weights, and graph algorithms are applied. Mishra et al. (2024) simulate random walks on a 3D graph to approximate CO2 plume spreading in a reservoir. Both Knudby and Carrera (2006) and Rizzo and de Barros (2017) demonstrate in 2D that shortest path algorithms approximate quite well the trajectory of the fastest particles in the plume and the drawdown signal.

The objective of this paper is to demonstrate how useful and efficient can graph-based approximations of flow and transport can be to reduce geological concept uncertainty in groundwater applications. To do so, we adapt the approach of Rizzo and de Barros (2017), improving its consistency with subsurface flow, and extending its application to a 3D case with increased complexity in terms of heterogeneous aquifer properties, by considering a faulted multi-heterogeneous-layer medium. In particular, rather than focusing solely on the best path between the source and a set of target nodes, we calculate the minimal

distance between the source and each node, resulting in a distance map. We compare this distance distribution to the distribution of cumulative mass passing through the outlet, to evaluate the accuracy of our model. We also assess the robustness of the approximation under the uncertainty of parameters controlling the heterogeneity of subsurface properties. In addition to measuring the performance of this new method for scenario selection, as compared to using more expensive physics-based numerical solvers, we provide a way to predict fault behavior a posteriori, based on field measurements.

The manuscript is organised as follows. The methodology employed is described in Sect. 2. It starts by introducing the synthetic experimental setting (Sect. 2.1) including a description of the medium heterogeneity and the necessary conditions to numerically solve flow and transport equations. Then, Sect. 2.2 presents how we approximate flow and contaminant transport using distance computation through graphs. Section 2.3 specifies the modalities for observing data from the physics-based model. Section 2.4 introduces metrics to allow the comparison between distance maps from graph computations and cumulative

mass maps. Section 2.5 shows to what extent this method can be applied to the selection of fault scenarios. Section 3 presents the general results, highlighting the correlation between the distance distributions and the distribution of cumulative masses,





and the effectiveness of the method for scenario selection. Finally, the conclusion and the possibilities for future experiments are discussed in Sect. 5.

## 2  Method

### 2.1  Experimental setting

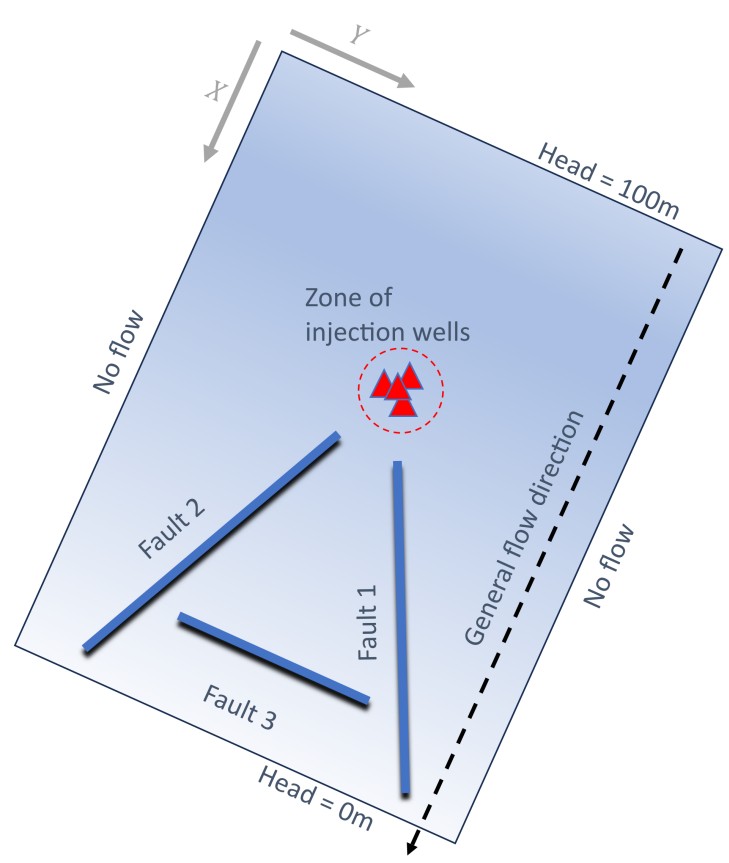

**Figure 1.** Geometry of the simulation. Faults planes are orthogonal to the x-y plane represented.

For this paper, we consider the following synthetic case, depicted in Fig. 1: a fault zone with three vertical faults and three geological units, each characterized by different heterogeneous property field parameterization. The flow propagates primarily in the $x$ direction, with the system's inlet and outlet maintained at constant head. A contaminant is continuously injected approximately in the middle of the model, and we study the transport of this contaminant until it exits the model at its

outlet face. Flow in a heterogeneous medium is modeled by two equations: Darcy's law and the advection-diffusion equation. These differential equations can be solved using a finite element solver. Faults influence the transport of the contaminant by locally altering the hydraulic conductivity. In this synthetic case, we assume that the faults can either increase or decrease the





conductivity by a factor of 100, so they can act either as a pathway (1) or a barrier (-1). Considering all possibilities, there are 8 possible fault scenarios, designated by a triplet $(f_1, f_2, f_3)$ belonging to $\{-1, 1\}^3$. The highly schematic geometry of the faults was chosen to maximize the variability of the plume depending on the behavior of the faults. We add further variability by testing 10 possible source positions, chosen randomly around a reference point with coordinates $x_s = 2000$, $y_s = 2500$, $z_s = 512.5$. This results in a total of 80 scenarios, numbered from 0 to 79, where the tens digit refers to their fault scenario and the units digit refers to the source position.

The model dimensions are $L_x = 7000$ m, $L_y = 5000$ m, and $L_z = 1000$ m. Spatial discretization is done in cells of size $\Delta x = 100$ m, $\Delta y = 100$ m and $\Delta z = 25$ m. The primary direction of flow is along the x-axis: the heads at the planes $x = 0$ and $x = 7000$ are constant and equal to 100 m and 0 m, respectively. The other boundaries of the model are constrained by zero flux. In our study, we assume a point source (one cell) that continuously injects a contaminant at a rate of 50,000 $m^3/d$ with a concentration of 100 units of mass per $m^3$. The soil consists of three geological units with average conductivity of 3.5e-5, 8e-4 and 2e-5 $m.s^{-1}$ respectively and constant porosity of 0.25 . The hydraulic log-conductivity (before the effects of faults) is modeled by a spatial random field (SRF) with a multi-Gaussian (MG) model, with a standard deviation of 0.4, 0.5 and 0.6 respectively and a correlation length of $8\Delta x, 4\Delta y$ and $2\Delta z$ respectively. The faults are modelled directly on the regular-grid voxel, so each fault occupies an ensemble of face connected voxels in the model. An example of sections of the hydraulic conductivity field is shown in Fig. 2.

## 2.2 Graph generation and Computation

Here we explain how the regular-grid discretization of an aquifer model can be converted into a graph for distance computation approximating flow and transport.

### 2.2.1 Graph generation

A graph $G(V, E)$ is defined as a pair comprising a set $V$ of vertices and a set $E$ of edges. Each edge $e \in E$ connects two vertices in V and may have an associated weight. In our study, the graphs are directed, and they always have an associated geometric dimension. Thus, for each edge $e$ connecting vertex $v_1$ to vertex $v_2$, we denote by the vector $\boldsymbol{e}$ the directed vector between the two corresponding points in 3D space. Lastly, a path is described as a sequence of vertices where each pair of consecutive vertices is linked by an edge.

The conductivity fields used by physics-based solvers like MODFLOW are discrete fields; here we work exclusively with regular grids of a bounded 3D space. To construct the graph, we choose the center of each cell in the discrete field mesh as vertices. Two vertices are connected by an edge if their respective cells share a face or a corner.

For an edge $e$ connecting two vertices $v_1$ and $v_2$, we can calculate an approximation of the hydraulic conductivity $K(e)$ along this edge using the harmonic mean:

$$K(e) = 2 \cdot (K(v_1)^{-1} + K(v_2)^{-1})^{-1} \tag{1}$$



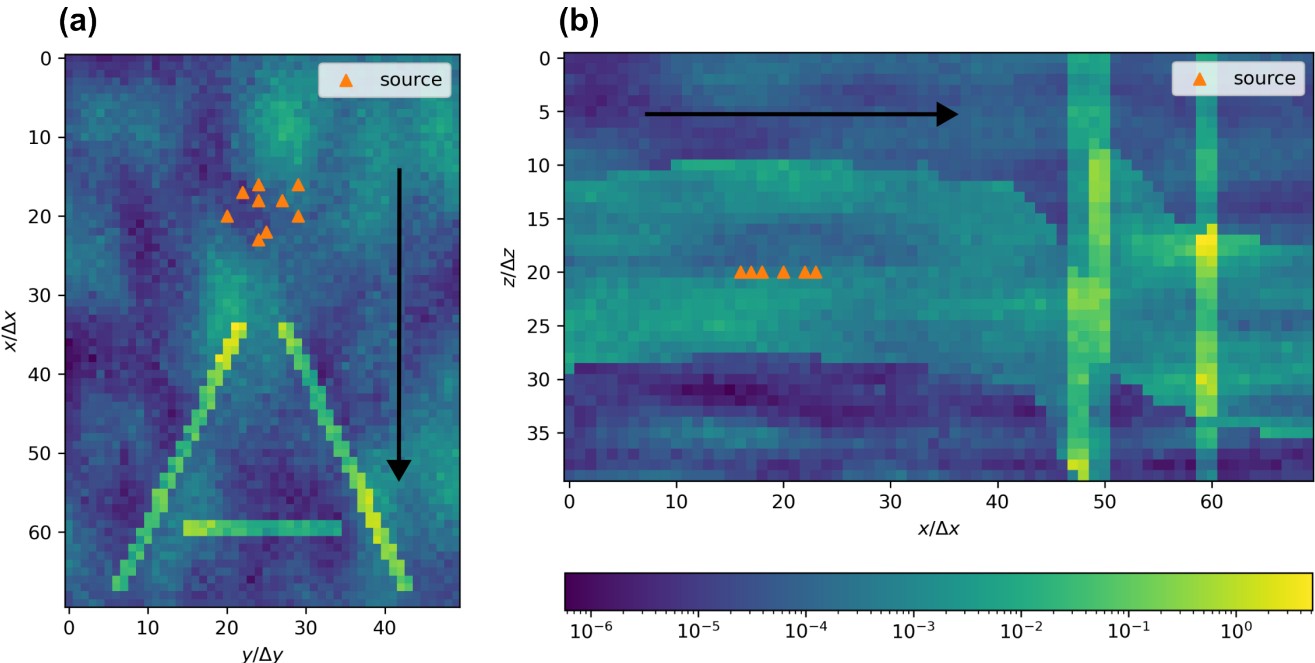

**Figure 2.** Sections of the hydraulic conductivity $(m/s)$ field for fault scenario $(1,1,1)$, with the faults increasing the conductivity. (a) : section with $z/\Delta z = 30$ (b) : section with $y/\Delta y = 15$ The possible injections wells are in orange. The black arrow represents the main direction of the flow. The axes are expressed in discretization units.

where $K(v_1)$ and $K(v_2)$ denote the hydraulic conductivity tensors in the respective cells of vertices $v_1$ and $v_2$.

For a given path $\Gamma$ within 3D space, hydraulic resistance is defined by the following formula (Rizzo and de Barros, 2017):

$$R_\Gamma = \int_\Gamma |K^{-1}(l) \cdot \boldsymbol{dl}|, \tag{2}$$

The concept of hydraulic resistance to groundwater flow is important because the fluid tends to follow paths of minimal resistance (Le Goc et al., 2010). Note also the similarity of this concept with that of electrical resistance. We can discretize this definition to apply it to our model. For a given oriented edge $e \in A$, its hydraulic resistance can be approximated by the

formula:

$$R_e = |K^{-1}(e) \cdot \boldsymbol{e}|, \tag{3}$$

Rizzo and de Barros (2017) use this value of hydraulic resistance for their modeling. In our case, in 3D and for a point source, we found that the results were more conclusive by adding a corrective factor to this formula in the form of a dot product, preventing paths from going "backwards". For a given edge $e \in E$, its weights $w_e$ is defined as:

$$w_e = \max(\boldsymbol{f_{dir}} \cdot \boldsymbol{e}, 0) \cdot R_e, \tag{4}$$





where $f_{dir}$ is the main direction of the flow.

Thus, we obtain a graph with exactly the same resolution as the original simulation space, with edge weights that accurately approximate the cost for the contaminant to traverse that edge.

### 2.2.2 Computation

The shortest path problem is a classic problem in graph theory. It has several variants, depending on the number of sources, targets, and the nature of the weights. In our case, we aim to find the shortest paths between a single source (the contaminant source) and all nodes in the last layer of the model. The algorithm of choice in this case is Dijkstra's algorithm (Dijkstra, 1959). The graph utilized is the one generated in section 2.2.1, with each edge $e$ being assigned the weight $R_e$.

Starting from the weighted and directed graph generated in the previous section, we aim to apply a shortest path algorithm

between the source and the graph nodes corresponding to the model outlet face (for which the hydraulic head is set to 0m on Fig. 1). Here, the source is a single point, and the model outlet face includes 2000 nodes. Rizzo and de Barros (2017) calculate only the shortest path between the source and the target set. In contrast, we calculate the minimum distance between the source and each node in the model outlet face. This process is not costly because, in general, Dijkstra's algorithm needs to compute all distances to obtain any particular one. We thus obtain a distance value for each vertex in the last layer, resulting

in a 2D array that can be visualized as an image. An example is provided in Figure 3 (b). In practice, we used the function "get_shortest_paths" from the Python igraph library (Csardi and Nepusz, 2005), which is compiled in C++. In the following content, the distance map returned by Dijkstra's algorithm is denoted as $I_d$.

### 2.3 Observation time

Equivalent simulations were performed with MODFLOW to compare the shortest paths with concentrations calculated numer-

ically. To make the comparison possible, an observation time for the simulation must be chosen. Indeed, while modeling the geological environment as a graph and calculating shortest paths does not depend on time, the model outlet face concentrations calculated by MODFLOW can vary considerably depending on the chosen observation time. The question is which value and at which observation time to compare with the values returned by the Dijkstra algorithm. Logically, one can expect that the shortest path algorithm will better approximate the path taken by the fastest particles of the fluid rather than the slowest ones.

Therefore, it seems reasonable to choose a relatively short observation time, a First Time of Arrival (FTA). Here, we define it as the time in the numerical simulation at the point when the cumulative mass that has passed through the last layer reaches 1% of the injected mass during the first time step, similarly as in Rizzo and de Barros (2017). For our observations on the last layer of the model, we have chosen to examine the cumulative mass that has passed through it since time $t = 0$ rather than the concentration, to be less sensitive to this observation time. An example is given in Fig. 3 (a). In what follows, the cumulative

mass map returned by MODFLOW at FTA is denoted as $I_m$.





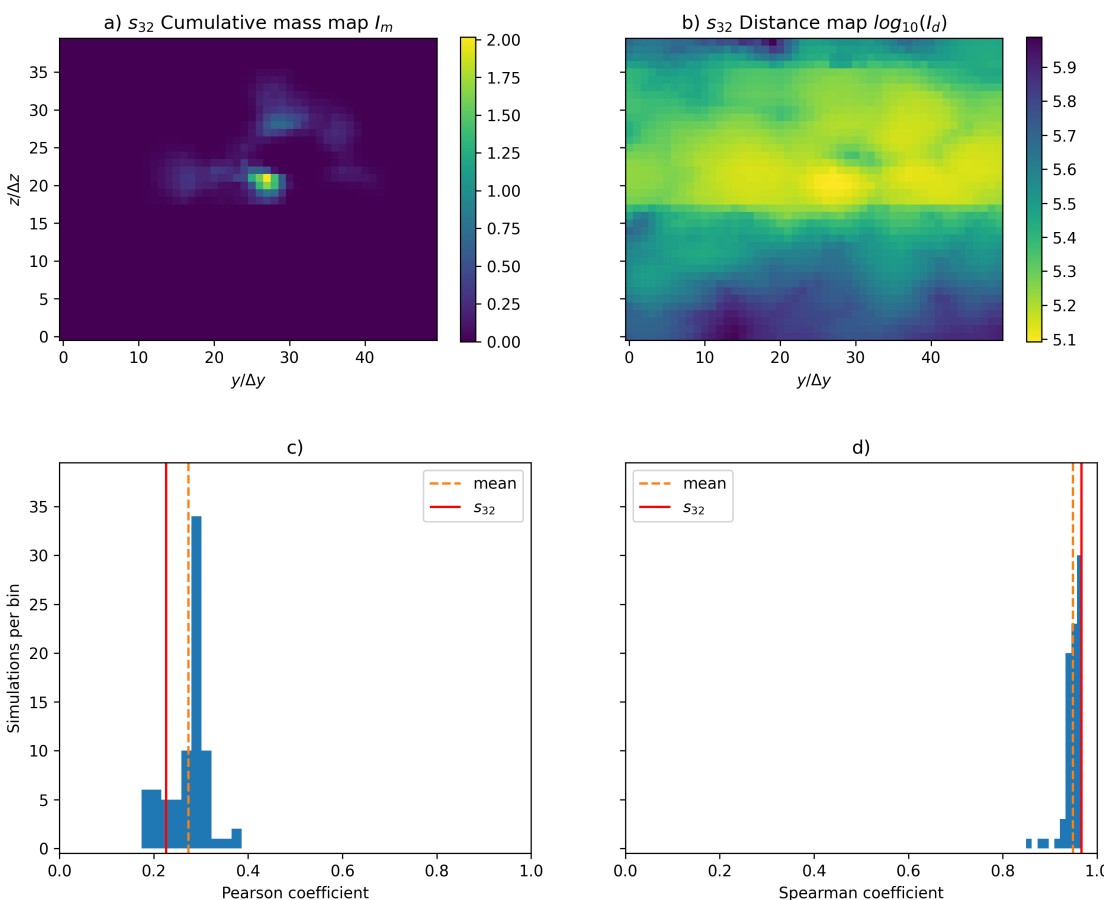

**Figure 3.** (a) : Map of the cumulative mass at FTA for scenario 32, (b) : Map of distances between the source and the model outlet face for scenario 32, (c) and (d) : Histogram and mean of two correlation coefficients between the negative of the distances and the cumulative mass over all the 80 scenarios.





## 2.4 Metrics

Comparing briefly the distances map and the cumulative mass map over the 80 cases (Fig. 3 (a) and (b)), one can see that the distributions look quite dissimilar. The histograms displaying Pearson and Spearman correlations between the two distributions have been calculated and can be found in Fig. 3 (c) and (d). The average values for these correlations are around 0.2 for Pearson

and above 0.9 for Spearman. This suggests that while there is a relatively weak correlation between the distributions themselves, the rankings of the pixels exhibit a strong correlation. What interests us more than the correlation between the two entire arrays are the pixels in $I_m$ with significant cumulative mass. We want to find a metric that spatially compare these pixels to the pixels in $I_d$ with low Dijkstra distances. This inspires the following method, represented in Fig. 4:

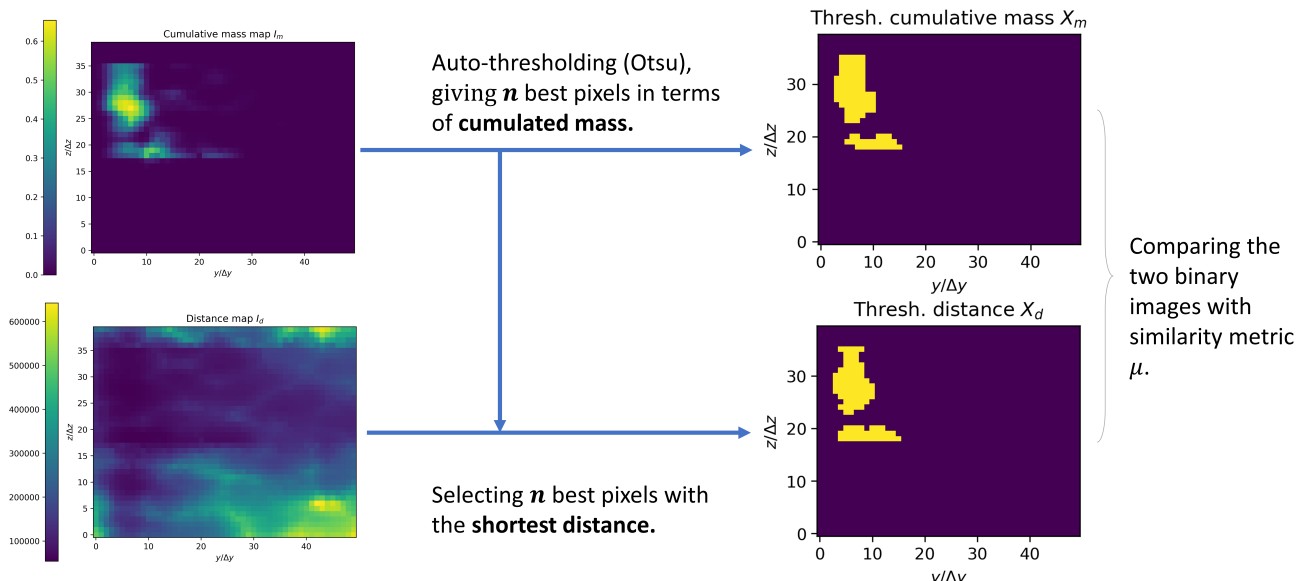

**Figure 4.** Method to compare cumulative mass and distance maps.

1. we identify pixels in $I_m$ where the cumulative mass exceeds a certain threshold, denoting them as significant concentra-
tion zones, defining a set of $n$ points $X_m$,

2. the $n$ pixels with the smallest distances are selected in $I_d$, defining a second set of points $X_d$,

3. for a certain similarity metric $\mu$, $\mu(X_m, X_d)$ is computed.





For step 1., the Otsu thresholding method is utilized (Otsu, 1979). This thresholding method minimizes the intra-class variance for a distribution. It has the strong advantage of being non-parametric and is considered a reference in computer graphics. For step 3. of comparing between the two sets of points, the problem is reduced to comparing two binary images, assigning label 1 to points in the sets of interest and 0 to others. Numerous metrics exist in the machine learning literature for segmentation problems. We have chosen to employ two complementary metrics: the Jaccard similarity index and the Normalized Wasserstein Distance.

The Jaccard index, also known as IoU (Intersection over Union) ratio, quantifies the similarity between two finite sample sets $A$ and $B$ as follows:

$$J(A,B) = \frac{|A \cap B|}{|A \cup B|} \tag{5}$$

In our context, the sets in question are the non-zero pixels $X_m$ and $X_d$ from each image. The Jaccard index is beneficial because it evaluates the overlap between the spots in both images and ranges from 0 (indicating total dissimilarity) to 1 (indicating total similarity). However, its limitations, as outlined in Wang et al. (2022), include a predisposition towards larger areas rather than smaller ones. In the latter, a single-pixel error might significantly impact the IoU ratio. Moreover, the index drops to zero with no overlap between the spots, failing to differentiate between various non-overlapping scenarios, including those where a spot's shape remains preserved despite translation.

Another valuable metric is the Wasserstein distance, or Earth Mover Distance , derived from optimal transport theory. This measure assesses the dissimilarity between two distributions or densities by calculating the 'cost' of transferring matter from one distribution to the other. The Wasserstein distance can vary depending on the underlying distance metric; in our study, we utilize the Euclidean distance, yielding the 2-Wasserstein Distance ($W_2$), which is the square root of the loss from the following optimization problem:

$$W_2^2(a,b) = \min_{\gamma \in \mathbb{R}_+^{m \times n}} \sum_{i,j} \gamma_{i,j} \|x_i - x_j'\|_2$$
$$\text{s.t.} \quad \gamma \mathbf{1} = \mathbf{a}$$
$$\gamma^T \mathbf{1} = \mathbf{b}$$
$$\gamma \geq 0$$

where $a$ and $b$ represent the sample weights, or in other words the mass distribution to be displaced, and $(x_i)_i$ and $(x_i')_i$ are the euclidean coordinates of the points from the two samples, respectively. The solution of the optimization problem $\gamma$ is the optimal transport matrix between the two samples.

In our case, we have two binary images. We can consider these two binary images as two 2D uniform distributions over the points with a value of 1, where each point with a value of 1 in the binary image has a value of 1/n in the distribution, ensuring





that this distribution is properly normalized. Thus, we take

$$a = b = \left( \tfrac{1}{n} \ldots \tfrac{1}{n} \right)^T$$


and $(x_i)_i$ and $(x'_i)_i$ as the 2D-coordinates of the elements of the sets $X_m$ and $X_d$ respectively.

Directly dealing with this distance can be challenging due to its dependence on the data type, including the sample size and the characteristic distance between samples, and because it does not scale between 0 and 1. An approach, as developed in Wang et al. (2022), introduces the Normalized Wasserstein Distance (NWD), which scales from 0 (indicating total dissimilarity) to 1

(indicating total similarity):

$$\text{NWD}(X_m, X_d) = \exp\left( -\frac{W_2(X_d, X_c)}{C} \right)$$

where C is "a constant closely related to the dataset" (Wang et al., 2022). C is chosen as the average standard deviation of the coordinates of the sets $X_m$ and $X_d$, calculated across multiple scenarios. The NWD has the advantage of better accounting for results that are merely translated, correlating closely with the distance between the centers of mass of the distributions (Lipp

and Vermeesch, 2023), but it has the disadvantage of overly penalizing cases where a pixel is very far from the center differs between the two images.

To mitigate this, we have calculated the arithmetic mean of the Jaccard Index and the NWD as a similirity index, denoting as $\mu$ :

$$\mu(X_m, X_d) = \frac{\text{NWD}(X_m, X_d) + J(X_m, X_d)}{2} \tag{7}$$

Thus, following the method above, we select the pixels of interest in both images $I_m$ and $I_d$, and we calculate their similarity index using the function $\mu$. Therefore, we define the function $\mu^*$, which performs all of this for two images $I_m$ and $I_d$ and returns their similarity:

$$\mu^*(I_m, I_d) = \mu(X_m, X_d) \tag{8}$$

## 2.5 Method of scenario selection

Suppose we are dealing with a geological setting where we know the conductivity field and the source position, and we have measurements of the cumulative fluid mass that has traversed the output layer up to the present time. Faults are present, but we are uncertain whether they behave as preferential path or barrier. Can we predict the nature of these faults using our shortest path method and our similarity metric $\mu$? To address this question, we aim to compare the similarity between binary images generated from MODFLOW (which we consider to be our ground truth or reference data) and those resulting from the shortest

path calculations, as described in the preceding sections.

For a known source position $j$ and conductivity field, let $S_j$ be the set of fault scenarios, and $n_f = |S|$ the number of possible fault scenarios. Let us denote for each scenario $s$, $I_m(s)$ and $I_d(s)$ respectively as the arrays of cumulative mass and distances. Two methods to predict the fault scenario are described in the following sections: one method selects a set of scenarios to reduce uncertainty, while the other assigns a probability to each fault for its behavior.





### 2.5.1 Scenarios selection

We are striving to develop a method to identify, from a discrete set of potential scenarios, which scenario aligns with the actual measurements of cumulative mass in the output layer. However, we have noted that despite the accuracy of the MODFLOW simulation, certain scenarios lack sufficient variability to be distinguished, especially when the fault that distinguishes them has minimal or no impact on the plume. Thus, given a reference scenario ($s_0 \in S_j$), we aim to devise a strategy (represented by a function $f$) to select a set $f(s_0)$ of scenarios (instead of one scenario) that includes our reference scenario $s_0$. Equivalently, this function would reject certain scenarios and thus reduce the uncertainty. This function should rely exclusively on the cumulative mass map $I_m(s_0)$ of $s_0$ and the set of distance maps from all scenarios $\{I_d(s), s \in S_j\}$. Ideally, we would like to find a function satisfying $f(s) = \{s\}$ for every scenario $s$, but as we said this is not always possible due to the low inter-scenario variability of the model and the approximations of our method based on shortest paths.

Thus, we define the success of a strategy $f$ applied to scenario $s$, denoted as $Y(f,s)$:

$$Y(f,s) = \mathbb{1}_{\{s \in f(s)\}}, \tag{9}$$

returning 1 if $s \in f(s)$, 0 otherwise. We also define $w(f,s)$, that is the number of scenarios selected by the strategy $f$ for the scenario $s$:

$$w(f,s) = |f(s)| \tag{10}$$

From these results, we can calculate, for a strategy $f$, the success rate $\bar{Y}(f)$ and the average number $\bar{w}(f)$ over all available 80 scenarios (for all possible sources). We would like to maximize success rate while keeping the average number of selected scenarios low enough to reject the maximum number of scenarios. Therefore, we seek a strategy with the highest possible ratio $\frac{\bar{Y}(f)}{\bar{w}(f)}$.

We may simply consider the strategy $h_k$, for $k \in \{1, \ldots, n_f\}$, which always randomly selects a subset of $S$ of size $k$. For this type of strategy, we obtain a linear response: $\bar{Y}(h_k) = \frac{k}{n_f}$ and $\bar{w}(s) = k$. This dummy strategy serves as a baseline for improvement; a good strategy should display a metric above this linear response.

An idea for a method is for a certain threshold $\lambda \in [0,1]$ to retain only the scenarios $(s_i)$ such that $\mu^*(I_m(s), I_d(s_i)) \geq \lambda$. We thus define the strategy $g_\lambda$, for $\lambda \in [0,1]$:

$$g_\lambda(s) = \{t \in S_j, \mu^*(I_m(s), I_d(t)) \geq \lambda\} \tag{11}$$

Another idea is to select all the scenarios with maximum similarity $I$. This defines the $u$ strategy:

$$u(s) = \operatorname*{argmax}_{j \in S_j} \Big( \mu^*(I_m(s), I_d(j)) \Big) \tag{12}$$

### 2.5.2 Prediction of each fault's behavior

Another way to learn more about the reference scenario is to attempt to predict the behavior of each individual fault rather than directly seeking to identify the correct scenario. For a given reference scenario $s_0$, we want to calculate a value interpreted as a





similarity to predict the behavior of its fault $i$ ($i \in \{1, 2, 3\}$ in our example). For a given scenario $s_0$, we define the binary value $F_i(s_0)$, which equals 1 if fault $i$ is a preferential path in scenario $s_0$, and -1 if it is a barrier.

The sum of similarities between the reference scenario $s_0$ and the scenarios where the fault $i$ behave as a path, divided by the sum of similarities to the reference scenario for all scenarios, returns a value between 0 and 1 that can be interpreted as a probability:

$$P(F_i(s_0) = 1) = \frac{\sum_{s \in S_j, F_i(s)=1} \mu^*(I_m(s_0), I_d(s))}{\sum_{s \in S_j} \mu^*(I_m(s_0), I_d(s))}. \tag{13}$$

Indeed, it ranges from 0 (if $s_0$ is very dissimilar to the scenarios with $i$ as a preferential path) to 1 (if $s_0$ is very similar to these scenarios). Similarly, the probably for the fault $i$ to behave as a barrier for scenario $s_0$ can be seen as this normalized sum :

$$P(F_i(s_0) = -1) = \frac{\sum_{s \in S_j, F_i(s)=-1} \mu^*(I_m(s_0), I_d(s))}{\sum_{s \in S_j} \mu^*(I_m(s_0), I_d(s))} \tag{14}$$

$$= 1 - P(F_i(s_0) = 1) \tag{15}$$

$$\tag{16}$$

This value ranges from 0 and 1 as well, allowing it to be interpreted as a probability. It enables us to predict the behavior of the fault, by rounding it to 0 or 1.

## 3    Results

### 3.1    General Graph approximation performances

The similarity index described in Sect. 2.4 has been applied to analyse the results of the 80 scenarios. A representative sample of the results can be found in Fig. 5, and the distribution of similarities is shown in Fig. 6. The mean and median similarity over all scenarios are respectively 0.62 and 0.74.

The similarity value is indicative since it was constructed from two different measures and thus requires some interpretation to decide if the approximation is 'good enough' or not. Across all results, we observe that the approximation of $X_m$ by $X_d$ is

acceptable when the similarity value is greater than 0.3. We can conclude that the distance map provides a good indication of where the cumulative mass will be significantly present.

Another important result is the comparison of the computational time between the graph-based method and the physics-based method. We conducted our calculations on our model as well as two others with coarser (Low Resolution) and finer (High Resolution) resolutions. For the Low resolution, discretization parameters $\Delta x$, $\Delta y$, and $\Delta z$ are multiplied by 2, and

divided by two for the High resolution, resulting in the cell volume being either multiplied or divided by 8. The computation times are presented in Table 1. We observe that for the method using graphs, generating the graph has a significantly higher cost than calculating the paths. Moreover, the graph generation followed by Dijkstra's calculation takes approximately 10 times less computational time than the MODFLOW simulation.



**Figure 5.** Computation of the similarity for 6 different scenarios. For each case, on the left side is the cumulative mass (Xm) at FTA from MODFLOW, to which an Otsu thresholding is applied. On the right side, the map of distances (Xd) is shown, thesholded with the same number of pixels as for cumulated mass map. The similarity values are shown on the top of each figure. The axes are expressed in discretization units.

## 3.2 Scenario selection illustration on two examples

To illustrate the previous methods on a concrete case, we choose scenario number 65 from our database (denoted as $s_{65}$), which has the source position 5 and corresponds to the fault scenario triplet $(1, -1, 1)$ (i.e., faults 1 and 3 are paths, and fault 2 is a barrier). Figure 7 shows the similarity between the cumulative mass map $I_m(s_{65})$ and each of the distance maps from all fault scenarios $I_d(s), s \in S_5$. We can see that two fault scenarios stand out distinctly, fault scenarios $(1, -1, -1)$ and $(1, -1, 1)$, thus



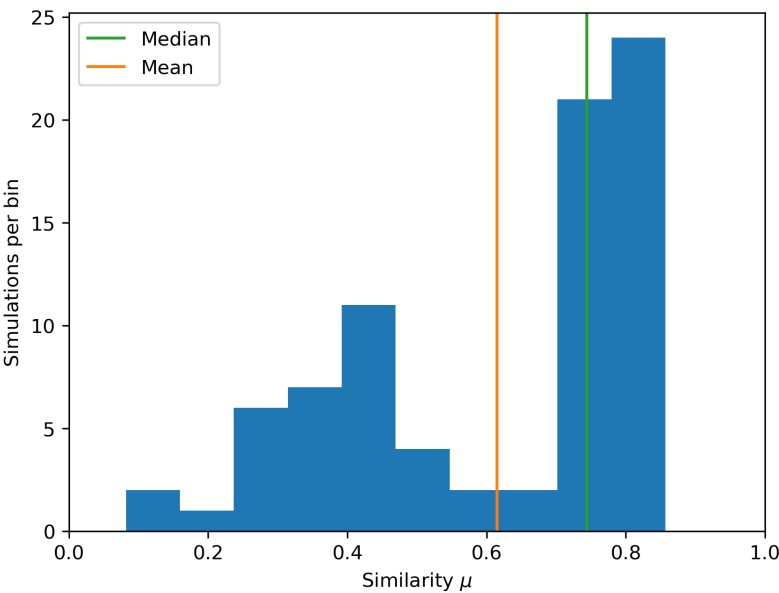

**Figure 6.** Histogram of the similarity index values over all 80 scenarios.

**Table 1.** Duration of each simulation for 3 different model resolutions. The physics-based simulation is conducted with MODFLOW, the graph-based one with the igraph library.

|  | Physics-based simulation | Graph Generation | Dijkstra computation |
|---|---|---|---|
| Low Resolution | 10.6s | 1.5s | 0.02s |
| Standard Resolution | 80s | 10s | 0.25s |
| High Resolution | 712s | 95s | 2.6s |

including the correct scenario. Therefore, with the strategies defined in Sect. 2.5.1 $g_\lambda$ (with any threshold between 0.2 and 0.75) or with the strategy $u$, we can clearly isolate these two scenarios from the rest, allowing us to reject 6 out of 8 fault scenarios. If we attempt to predict the faults individually (as in Sect. 2.5.2), we obtain the probabilities in the second row of the Table 2. The prediction is accurate for faults 1 and 2, but for fault 3, the probability is very close to 0.5, not allowing any conclusion. For scenario 65, we see that both approaches allow for the clear identification of the nature of two out of three faults.

Now, let us consider scenario $s_{12}$ from our database, which has the source position 2 and corresponds to the fault scenario $(-1, -1, 1)$. Looking at Fig. 7, which shows the similarity between the cumulative mass map $I_m(s_{12})$ and each of the distance maps from all fault scenarios $I_d(s), s \in S_2$, we can see that it is less clear here. Even if the correct fault scenario has the highest cross similarity, the difference with the others is not substantial enough to make a confident prediction. Using the second method and looking at each fault individually, we obtain the probabilities in third row of Table 2. While the prediction for the

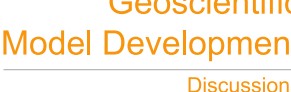
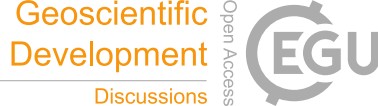

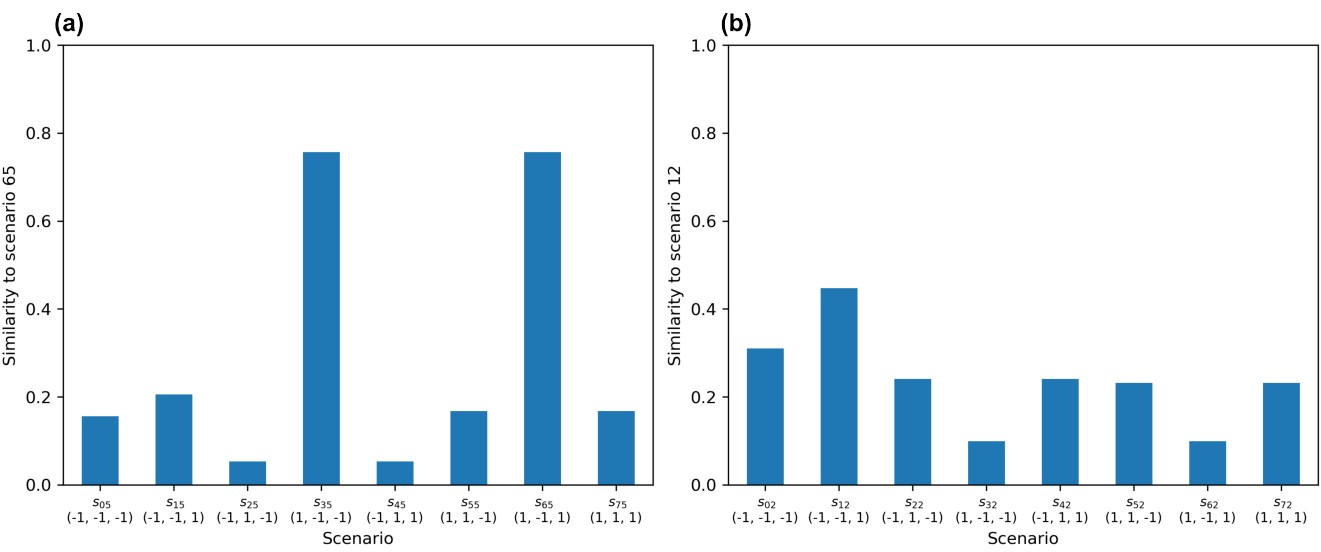

**Figure 7.** Cross similarity for scenarios 65 (a) and 12 (b).

**Table 2.** Probabilities $P(F_i(s) = 1)$ for each fault being a preferential path for scenario 22 and scenario 70.

| Scenario \ Fault id | Fault 1 | Fault 2 | Fault 3 |
|---|---|---|---|
| $s_{65}$ | 0.80 | 0.19 | 0.51 |
| $s_{12}$ | 0.35 | 0.49 | 0.53 |

first fault is clearly predicted (correctly) as behaving as a path, the prediction is poor for faults 2 and 3, with probabilities slightly

below or above 0.5. Thus, for this scenario, the results are less favorable, with only one fault being confidently identified.

### 3.3   Scenario selection overall results

The results of Success rate $\bar{Y}$ as a function of the average number of selected scenarios $\bar{w}$ are presented in Fig. 8. It is evident that all data points lie significantly above the baseline curve of the $h_k$ functions. Specifically, selecting the $g_\lambda$ function for $\lambda = 0.5$ yields a precision of $\bar{Y} = 0.8$ and an average number of selected scenarios $\bar{w} = 2$, which can be interpreted as a

confidence of 80% to select the right scenario when selecting the 2 best scenarios. Close results are obtained with the $u$ function. This shows that with this method we are able to confidently reject a good portion of the scenarios. Using the probabilities calculated in Equation 13, we can then calculate the recall and precision for each fault in predicting its behavior. Because there are two possible classes (barrier or path), recall and precision are calculated for both classes. The results are shown in Fig. 9 with blue markers. Also in Fig. 9, the recall and precision scores obtained using the cumulative mass results from MODFLOW

from start to finish are shown with orange markers. The fact that these precision and recall are not equal to 1 demonstrates the

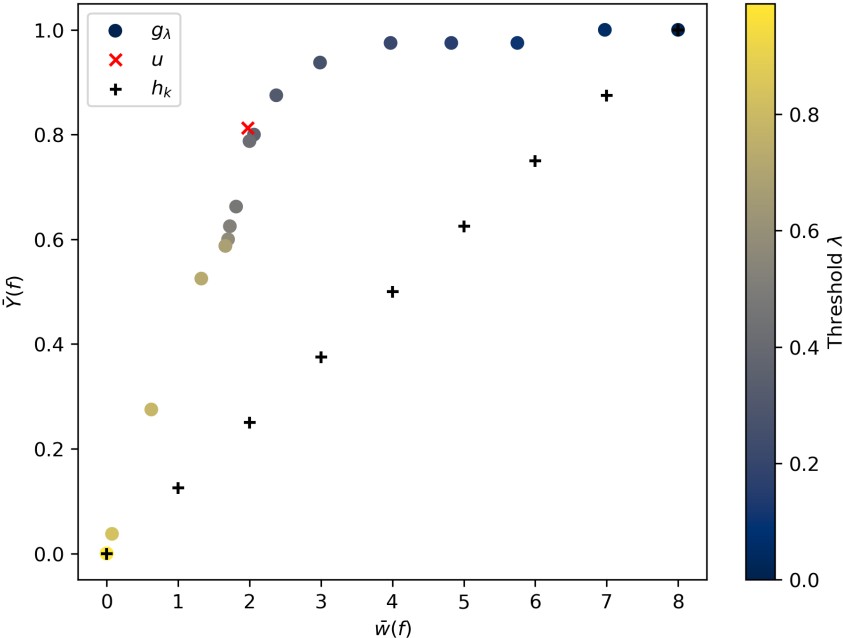

**Figure 8.** Results of scenario identification for different selection functions. Each point corresponds to one strategy $f$, and its coordinates corresponds to the average number of scenarios retained $\bar{w}(f)$ and the success rate $\bar{Y}(f)$, computed over all 80 cases. The black cross markers refer to the dummy strategies $h_k$, selecting a constant number of random scenarios. The dot markers refer to the strategies $g_\lambda$, retaining the scenarios with a cross similarity over the threshold $\lambda$, their color corresponding to the value of $\lambda$ according to the colorbar on the right. Finally, the red cross marker refers to the strategy $u$, selecting the scenarios with the maximal cross similarity. We can notice that both strategies $g_\lambda$ and $u$ are above the line of the random strategies $h_k$.

inherent lack of variability in the data, i.e. there exists ambiguity between scenarios that cannot be resolved when using the physic based solver. Even with perfect measurement, we cannot determine the nature of each fault with certainty a posteriori.

We can make the general observation that the results from the graph-based models are within the range of the results from the physics-based solver. Notably, for Fault 2, the graph-based model even outperforms the physics-based one in predicting its
behavior. This is because the graph method is highly sensitive to the presence or absence of paths with very high conductivity. Conversely, for Fault 3 (the transverse fault), the results are significantly worse. This is because the Dijkstra paths are minimally influenced by the nature of Fault 3 due to its geometry: whether it acts as a preferential pathway or a barrier, it only adds a constant to the length of all the paths.

## 4 Discussion

This study has confirmed and extended the findings of Rizzo and de Barros (2017) by successfully demonstrating the effectiveness of graph-based methods in approximating contaminant transport in 3D subsurface environments with faults. Using



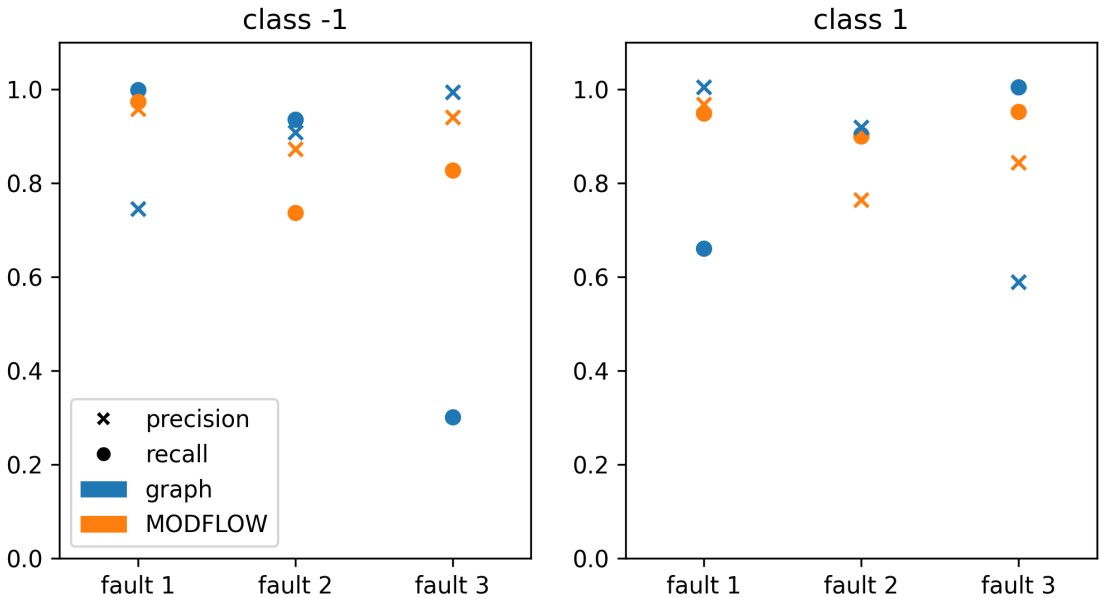

**Figure 9.** Precision and recall for the classification for each fault, each method, and each class. (a) : class -1 (barrier), (b) : class 1 (path)

a graph modelisation very similar to the one of Rizzo and de Barros (2017), but embedding a few improvements, we have shown that not only the shortest path, but the whole distance map generated by Dijkstra's algorithm between the source and the model's outlet is rank-correlated with the distribution of cumulative masses flowing through the outlet. The proposed metric,

combining both Jaccard index and Wasserstein distance, and used to compare the graph based distances with the cumulative mass, is effective to compare binary images and exibits fairly good spatial similarity between the two maps.

In addition to the model presented in Sect. 2.1, we tested our method in the absence of faults by varying the multi-Gaussian field. The results are presented in Appendix A. We first verified that our results align with those of Rizzo and de Barros (2017). We also studied the uncertainty of the minimal distance point of the outlet and compared it with that of the maximum

cumulative mass point. We demonstrated that the uncertainties were comparable and followed similar trends for different field parameters.

These results suggest the potential use of graph-based methods as a proxy for groundwater flow simulation. In particular, when traditional methods are too costly to implement and when the sought-after information is less about the contaminant concentration values and more about its location on a control plane. This is relevant for scenario selection, which can be

achieved by comparing the locations of contaminants at the outlet. Our experiment described in Sect. 2.5 allowed us to asses the use of a graph-based method in fault scenarios selection. By comparing the similarity between the cumulative mass result of a reference scenario and the graph simulations, we can either reject a significant number of scenarios to reduce uncertainty



or calculate a fault-by-fault probability of increasing/decreasing the conductivity. For two of the three faults studied, our results are close to those obtained with MODFLOW.

330 However, several questions and challenges related to the use of graph-based methods remain unresolved after this study. It is still necessary to explore the impact of the chosen observation time for the physical data, the possibility of 3D visualization of the shortest paths, and to test other graph algorithms for approximating groundwater flow. Additionally, the difficulty in determining a thresholding method for the distance seems to compromise the possibility of completely replacing physics-based methods. All these questions are detailed in the following paragraphs.

335 An aspect to consider is the attention given to the observation time. As mentioned in Sect. 2.3, we chose to perform all our measurements at the First Time of Arrival (FTA). While we use a percentage approach to determine the FTA, an alternative could be to use a deconvolution approach (Luo and Cirpka, 2008), potentially at greater computing expenses. Then, with our dataset, the time at which the distribution of cumulative mass is closest to the distribution of distances. However, it would be necessary to study the quality of the approximation at other observation times as well.

340 Additionally, it would be interesting to test other graph algorithms to approximate groundwater flow. In particular, the minimum-cost flow problem Ahuja et al. (1993) could be useful if it can be properly defined in this context. Specifically, it would be necessary to find a geological value to associate with the notion of capacity, knowing that hydraulic resistance can be used to represent the cost.

 With this graph-based method, we can hope for a true 3D visualization of the plume shape, rather than just the distance
345 distribution at the outlet. We have conducted some preliminary tests in this direction. The initial idea was to recalculate the distances between the source and each orthogonal section of the graph using Dijkstra's algorithm, rather than just the final section. However, this method was unsuccessful due to the lack of consistency in the distance distribution between different sections. A more successful idea was to calculate the number of paths passing through each node in the 3D mesh to identify the most visited nodes. Preliminary figures are presented in Appendix C. A more quantitative study such as comparing results
350 with streamline-based approaches would be necessary.

 Finally, there are two paths open to make the graph-based method fully independent from the physics-based results. The first would be to find a thresholding method to distinguish the pixels of interest solely based on their distance. We attempted this in Appendix B, but our results were mixed. The second, more ambitious method would be to find a function $\Phi$ that transforms the distance distribution $I_d$ into an estimate of the cumulative mass $\hat{I}_{mf} = \Phi(I_d)$. Machine learning approaches could be
355 considered for this. Developing a truly independent method could significantly reduce computation time, as graph generation and Dijkstra's calculation are 10 times less costly than a physics-based simulation.

 The investigation of the use of graph structures as proxies for geological processes extends beyond the hydrogeological application proposed here. Montsion et al. (2024) used Dijkstra distances as proxies for the non-Euclidean distance in 2D between geological features, by assigning weights to edges based on estimated flow properties, and these distances were in
360 turn used as part of a mineral prospectivity analysis. In the context of building 3D geological models, Graph Neural Networks are being used a framework for understanding relationships between observations (Hillier et al., 2021, 2023). In both cases the





possibilities for constraining the modelling results with knowledge graphs that share similar architectures (Enkhsaikhan et al., 2021) provides the potential for mapping specific local knowledge onto larger poorly understood regions.

## 5 Conclusions

GraphFlow allows for the calculation of Dijkstra paths to generate a distance map for the last layer of the model. We have demonstrated, by developing an appropriate similarity measure, that for a synthetic case involving a fault zone, these distance maps are highly rank-correlated (average Spearman coefficient of 0.9) with the distribution of cumulative masses at the First Time of Arrival (FTA). Moreover, the spatial similarity of the pixels of interest is high (0.62 on average for our similarity measure).

This result has enabled us to use this model for scenario selection. For 8 different fault scenarios, comparing their distance maps significantly reduces uncertainty by selecting a few plausible scenarios with confidence.

Several challenges remain in finding other applications for this method. The main challenge is in making the model independent of physics-based results: specifically, finding a threshold based solely on distance to distinguish between pixels of interest and pixels with negligible cumulative mass.

## Appendix A: Validation of the Graph-Based Approximation Method in a Heterogeneous Environment Without Faults

We also tested our graph-based approximation method in a heterogeneous environment without faults. We used the exact same parameters, but instead of testing variability according to fault behavior, we simulated 50 multi-Gaussian realizations for each geological unit, resulting in 50 different scenarios. There is only one source position with coordinates $x_s = 1050$, $y_s = 2550$, $z_s = 512.5$.

As in the main body of the paper, the distribution of similarity was calculated, with the mean and median being 0.37 and 0.38, respectively. These results, shown in Fig.A1, are significantly lower but still acceptable (above the qualitative threshold of 0.3). This can be explained by the absence of very high conductivity paths (or very low conductivity barriers), which the graph approximates quite well.

For these simulations, we also found it interesting to study the sensitivity of the groundwater flow simulation results to the parameters of the multi-Gaussian hydraulic conductivity field. This has already been tested in numerous papers for PDE-based methods only.

Cao et al. (2018) show that the characteristic size of the plume for a 2D simulation, as well as its variance (its uncertainty), increase when the field variance $\sigma$ increases, and also when the correlation length $\lambda$ increases. Srzic et al. (2013) also demonstrate that as the heterogeneity of the field increases, the uncertainty about the center of the plume increases as well. We would

like to see if the results from the shortest paths method exhibit similar behavior in response to parameter changes.

Starting from reference values for the standard deviation $\sigma_0$ and the correlation length $\lambda_0$, we successively apply a factor to vary both parameters. The variable we will focus on is the standard deviation of the position of the point of maximum





cumulative mass at the FTA (respectively, the point of minimal distance). For each standard deviation $\sigma$ and correlation length $\lambda$, we generated 50 realizations of the MG field and calculated the standard deviation of the coordinates of the point of

maximum cumulative mass at the FTA (respectively, the point of minimal distance). This represents the uncertainty of the result for fixed parameters standard deviation $\sigma$ and correlation length $\lambda$, considering that the exact structure of the conductivity field is often unknown. By decomposing the results on the $y$ and $z$ axes, we can visualize the results in figure A2. We can observe that in all cases, the results from Dijkstra's algorithm follow the trends of the MODFLOW results. Moreover, these trends are consistent with previously observed results in the literature: as the correlation length and standard deviation increase,

the uncertainty also increases. We can also notice the standard deviations from Dijkstra's algorithm are either equal to or significantly greater than those from MODFLOW. This means that the uncertainty related to the structure of the conductivity field is not underestimated by the Dijkstra's method.

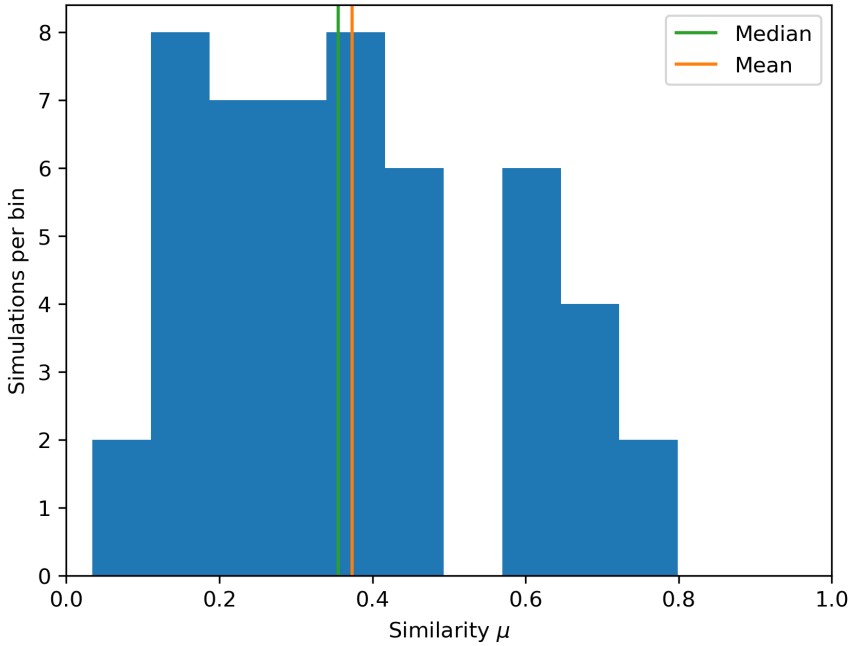

**Figure A1.** Histogram of the similarity index values over all 50 scenarios.

## Appendix B: Thresholding Methods for Identifying Significant Points Based on Distance

In Sect. 3, we observed the effectiveness of similarity: for a given number $n$ of pixels (corresponding to the number of pixels

where the cumulative mass at FTA is significant), we compared the set of $n$ pixels with the highest cumulative mass at FTA $X_m$ with the set of $n$ pixels with the smallest distance $X_d$. However, even with this knowledge, without physics-based data $I_m$, there is no straightforward way to determine which points of $I_d$ should be retained as locations where the contaminant





**Figure A2.** Standard deviation of point of maximum cumulative mass coordinates (resp. point of shortest graph distance coordinates) for simulations with MODFLOW, in blue (resp. with graph method, in orange) as a function of the correlation length of the conductivity field.

is present in significant quantities, based solely on the ranking of points according to their distance. For instance, we cannot predict whether the cumulative mass is uniform throughout the entire last layer or highly localized. Therefore, we aim to
automatically determine, using the distribution of distances, a threshold to distinguish between significant and other points, returning an estimation of the area $\hat{I}_{dk}$ where the contaminant is significant. The Otsu algorithm does not work well directly on distances array $I_d$ because the distribution is not suitable for it. By examining the distributions of several scenarios (see Fig. B1 (a) and (b)), we observe the presence of a peak, typically close to the minimum distance. Empirically, a correct threshold value consistently lies before this peak.





An attempt we made was to apply an Otsu thresholding to the signal before this peak. It's even possible to use multi-class Otsu thresholding to estimate different cumulative mass zones. The results are mixed, and some examples are shown in Fig. B1 (c) and (d). Often, our auto-thresholding attempt overestimates the area of interest.

**Appendix C: 3D Visualization of Dijkstra Pathways**

For each vertex, we aim to count the number of Dijkstra paths that pass through these nodes. Using the notations from section
2.2.1, and calling $(\pi_1, ..., \pi_{2000})$ the set of oriented paths calculated by Dijkstra's algorithm between the source and the 2000 nodes of the model outlet face, we define the number of paths passing through a vertex $v \in V$ as $n^*(v)$:

$$n^*(v) = \sum_{i \in \{1, ..., n\}} \mathbb{1}_{v \in \pi_i}. \tag{C1}$$

where $\mathbb{1}_{\{v \in \pi_i\}}$ is an indicator function that equals 1 if the vertex $v$ belongs to the path $\pi_i$, and 0 otherwise.

In practice, if we consider all the paths between the source and the last layer, we end up with nodes having a high $n^*$ value,
but these do not accurately correspond to the actual flow paths of the contaminant. This occurs because arrival points that are very far away or even at an infinite distance (in the sense of Dijkstra) from the source are counted, meaning the contaminant has no chance of reaching them. Thus, we realized that restricting the number of nodes to $m$ by selecting only the $m$ closest nodes (in the sense of Dijkstra) to the source yielded better results. For the examples, we arbitrarily chose $m = 200$, but this parameter warrants further exploration. Some examples of this method are shown in Figure C1.

*Code and data availability.*  The code to approximate groundwater flow and transport simulations via graph and reproduce the illustration examples with a set of illustrative notebooks are available at https://doi.org/10.5281/zenodo.13328938 (Moracchini and Pirot, 2024) as the v1.0 release of the GraphFlow GitHub repository under the MIT license.

*Author contributions.*  LM and GP framed the workflow of the proposed approach and developed the Python code and conducted the synthetic experiments and analysis. MJ participated in framing the idea. MJ and KB contributed to some parts of the code development. LM wrote the
core parts of the manuscript. All co-authors contributed to essential discussions, to the redaction and review of the manuscripts.

*Competing interests.*  The authors have declared that none of them has any competing interests.

*Acknowledgements.*  ChatGPT-4 was used to review some sections of this paper for English-language accuracy.





This work is supported by the ARC-funded Loop: Three-dimensional Bayesian Modelling of Geological and Geophysical data (LP210301239) and by the Mineral Exploration Cooperative Research Centre whose activities are funded by the Australian Government's Cooperative Research Centre Programme. This is MinEx CRC Document 2024/***.





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



**Figure B1.** (a) and (b) : Densities of the distance for two different simulations. The densities have been computed with a gaussian Kernel. The presence of a peak close to the shortest distance is to be noticed. (c) and (d) : Two different cases and their corresponding estimated thresholding on the distances. In both cases, the similarity is already quite good ($> 0.5$). The colorbars on the right refer to the discrete classes after the otsu thresholding, it is not meant to approximate the cumulative mass values. The axes are expressed in discretization units.







**Figure C1.** Visualisation for two scenarios of the most visited nodes $n^*$ and the concentration $C$ at FTA.