# Peer review of "GraphFlow v1.0: approximating groundwater contaminant transport with graph-based methods - an application to fault scenario selection"

_Geoscientific Model Development, 2024_

## Author Comment (AC1)

Reviewer comments (RC1) in black and answers in blue

**Positive Aspects**

- The graph-based approach simplifies complex geological models and reduces the computational costs.

- Distance map provides information about the potential pathways of contaminant transport.

- A new similarity measure used to compare the distance map to the cumulative mass distribution.

Thanks for your positive assessment of this work.

**General Comments**

- The term "groundwater" is often associated with specific subsurface conditions and flow regimes. While the principles of flow and transport in porous media can be applied to groundwater systems, the broader context of the study seems to be more general. It's important to use more accurate and inclusive terminology to avoid potential misunderstandings, a suggestion could be to use *porous media*.

  Our work is motivated by groundwater applications, as stated in the introduction. It is also illustrated with a groundwater synthetic case. While such work could have more general applications to flow and transport in porous media, it has not been tested. Nonetheless it could be tested and applied in other setting. This will be mentioned in the discussion at the revision of the manuscript.

- Including fault scenarios might seem unnecessary if the method doesn't perform well for cases without faults, as Appendix A shows.

    o Justify the Fault Scenarios: If the fault scenarios are crucial for real-world applications, provide stronger justification. Perhaps there are specific geological settings where faults significantly impact flow and transport.

    o Under this specific scenario, explore the limitations of the graph-based approach to justify the range of the metric that is considered acceptable.

    o Appendix A needs to include details of parametrization for the MODFLOW simulation.

  Exploring conceptual uncertainty such as different fault scenarios is a key motivation of our work, as highlighted in the introduction. As explained in Appendix A, the absence of very high conductivity paths (or very low conductivity barriers), which the graph approximates quite well can explain the mitigated performance of a graph-based approach in a multi-Gaussian setting.

> So, the use of the method is particularly interesting to tests scenarios displaying different types of hydraulic conductivity contrasts or pathways. This explanation will be added to the discussion at the revision of the manuscript. In Appendix A, we already state that the same parameters are used but we could precise 'same flow and transport boundary conditions'.

- The method still relies on a 3D simulation (MODFLOW) to generate the "ground truth" against which the graph-based method is compared. This limits the method's independence and its potential for significant computational savings. While the graph-based method can provide a quick and potentially accurate approximation, perhaps consider validation with simplified Analytical Solutions, Sensitivity Analysis or Machine Learning techniques. This would provide a more rigorous comparison without relying on numerical simulations.

    > The generation of a synthetic "ground truth" (here using MODFLOW) is indeed necessary to test our approach. In a real case study, is seems reasonable to have monitoring wells at the outlet of the model boundaries, such as to interpolate cumulated mass at the outlet. It would not require the use of flow and transport simulations.

- Similarity measure: A similarity coefficient of 0.3 might seem low, especially considering that a perfect match would be 1.0. While a higher similarity coefficient would be ideal, a value of 0.3 can still be considered reasonable but needs to be explicitly acknowledged, especially given the complexity of the problem. The authors should provide a detailed discussion of the factors influencing the similarity coefficient and explain why this value is acceptable in the context of their study. Additionally, the authors could explore ways to improve the accuracy of the graph-based method, such as refining the graph construction by experimenting with different graph configurations to capture the underlying geological features better.

    > Lines 263 to 266, we explain how we set this threshold of 0.3. Indeed, the proposed similarity metric is very sensitive to slight changes: a small shift both decreases the Wasserstein component of the similarity and decreases the Jaccard index. The advantage and drawback of each component are given in section 2.4 and justify the proposed metric. These additional explanations and a reminder of the dynamic of each component of the metric will be added to the discussion at the revision of the manuscript.

- A comprehensive evaluation of the graph-based method requires a clear understanding of the underlying physics-based model, including its setup and initial conditions. The authors should provide a detailed description of the MODFLOW simulations, including:
    - Model Domain: The spatial extent and discretization of the model domain.
    - Hydrogeological Properties: The values assigned to hydraulic conductivity, porosity, and other relevant parameters.
    - Boundary Conditions: The types of boundary conditions applied to the model boundaries.

    o Initial Conditions: The initial distribution of hydraulic head and contaminant concentration.

   All these characteristics are already described in section 2.1. Figure 2 illustrates sections of hydraulic conductivity for one of the fault scenarios. All scenarios are provided along with the code to generate them as explained in the Code and data availability section (line 430).

- Comparing a single MODFLOW scenario to multiple graph-based scenarios can be misleading, as it doesn't directly assess the accuracy of each individual graph-based scenario. A more appropriate approach would be to compare each corresponding pair of scenarios.

   As stated in lines 260-261, "The similarity index described in Sect. 2.4 has been applied to analyse the results of the 80 scenarios. A representative sample of the results can be found in Fig. 5, and the distribution of similarities is shown in Fig. 6." It means that for each of the 80 scenarios, we compute a similarity index (illustrated in Fig 5) between the MODFLOW cumulative mass map and the GRAPHFLOW distance map. The histogram of the 80 resulting similarity indices is displayed in Fig. 6.

   Regarding scenario selection, as explained in section 2.5.1, for a given contaminant source position all pairs of scenarios are tested against each other in the scenario selection. The success rate and number of scenarios are averaged across the 80 possible scenarios. We should precise that the average is over all pairs of scenarios and all contaminant sources.

   This will be added at the revision of the manuscript.

- The paper should be understandable to a broad audience without requiring extensive external references. Consider providing a brief explanation of the algorithms used:
  - o Dijkstra's Algorithm
  - o Other Algorithms (Jaccard dissimilarity, Wasserstein distance,Otsu thresholding)

   We already provide explanations of the Jaccard dissimilarity and Wasserstein distance in section 2.4 as we combine them to propose a new metric. For Dijkstra's algorithm and Otsu Thresholding, simple sentences (around lines 117 and 154 respectively) already describe the purpose of these methods, which is sufficient for the reader.

**Specific Comments**

- **Abstract:**

[2] The phrase "large-scale structural features" could be more specific. Explicitly mention geological features: "large-scale geological features, such as faults, fractures, and stratigraphic variations" and their standard scales compared to domain extension.

We will modify the text as suggested at the revision of the manuscript.

- **Introduction:**

[42-43] The paper should clearly state how the methodology " improves the consistency for subsurface flow". The author should provide a more precise explanation of why faults are relevant for contaminant transport in porous media. The manuscript should provide a deeper analysis of the role of heterogeneity within the graph-based approach.

The previous work from Rizzo and de Barros (2017) is limited to 2D multi-Gaussian heterogeneous medium and compares the graph approximation with results from particle tracking. Here we go one step further by integrating general flow direction information and by doing a comparison with flow and transport simulations (thus aiming at improving the consistency with subsurface flow). We will add these precisions at the revision of the manuscript.

[47] Consider addressing the role of heterogeneity in the main body of the manuscript.

I am not sure what this comment means. I agree that we should talk more about the role of heterogeneity in the first paragraph of the introduction. We will add these precisions at the revision of the manuscript and make sure that it is discussed properly with respect to application of the proposed approach when mentioning scenario selection.

- **Method:**

[60] Figure 1. There are no dimensions indicated in the figure. Is there a reason for the orientation of the scheme?

We will reorient the scheme and add dimensions as in figure 2 at the revision of the manuscript.

[70-73] The description of the experimental setting should be more specific about the position of the source points relative to the grid size. The authors indicate only one coordinate point; it is unclear where the random 10 positions fall on the modeling grid.

The 10 positions are displayed on Figure 2. We will add a reference to the figure in the revised manuscript.

[75-80] This section should also address how the authors evaluate the role of heterogeneity for the simulation domain for the different subsurface properties, as this section indicates a variability in the behavior of the faults but does not answer the effect of the hydraulic conductivity or porosity for this approach. Appendix A should be referenced here.

For each scenario, each geological unit is a stochastic multi-Gaussian SRF realization whose parameters are described lines 78 to 81. We will add this precision at the revision of the manuscript.

[98] Figure 2 shows the hydraulic conductivity values of one scenario. The color bar should be properly labeled, and the formatting of the relative position of the two plots needs to be adjusted.

The label of the colorbar will be added at the revision of the manuscript.

[100] Equation 2. This equation needs to be properly referenced and described in the text. The variables are not defined.

$R_\gamma$ is the hydraulic resistance along the path gamma. For each point $l$ on gamma (where $l$ is a dummy variable), we calculate the absolute value of the scalar product between the inverse of the conductivity tensor at point $l$, $K^{-1}(l)$, and the infinitesimal distance $dl$. We then compute the integral of this value over the path gamma.

This will be added at the revision of the manuscript.

[105] Equation 3. This equation needs to be properly referenced and described in the text.

It is the scalar product between the inverse of the hydraulic conductivity simplified tensor $[k_{xx}, k_{yy}, k_{zz}]$ and the oriented edge $[e_x, e_y, e_z]$. This will be added at the revision of the manuscript.

[126] is the function "get_shortest_paths" the same as the Dijkstra algorithm?

Yes, it is the implementation of Dijkstra algorithm used here.

[140] Figure 3. At this stage of the reading, it is still not clear what s32 is. The figure needs quality improvement. Include units for the color bars. Figures c and d should be moved further down as it is not clear at this point what they mean, and they are not formatted properly. Labels for figures c and d should indicate the modeling framework

used (MODFLOW, GRAPHFLOW). Furthermore, the choice of histogram plot to compare the output of 80 simulations using the new methodology compared to one single scenario using MODFLOW is confusing as it does not indicate the performance of each simulation against its corresponding physics-based.

S32 is scenario 32 (numbering is explained lines 73 & 74). Cumulative mass, computed from MODFLOW outputs, is in units of mass per cubic meter (as define line 78) and distances, computed from GRAPHFLOW outputs, are homogeneous to a length in meters, but are not meters as the length through the graph is the product of weighted lengths, so we do not prefer to add a unit to the colorbar of this subfigure, as it could be misleading for the readeran d) display the correlation coefficients between cumulative mass and graph distance.

- **Metrics**

[148] Figure 4 needs to improve its quality. Some recommendations: use the same font size of the plots and add labels to the color bars and units of measure. Adjust formatting. Since this is a workflow of the proposed metric, use more descriptive texts next to the figures.

We agree with these formatting recommendations and will implement them at the revision of the manuscript.

[178] Variables have different formatting than the previous equation. 2-Wassertein Distance (W2) needs to be numbered.

We have checked the formatting and all 'b's should be bold and italic. The equations formatting, as well as the equations numbering, will be updated at the revision of the manuscript.

- **Method of scenario selection**

[205-214] This section seems to address a different problem: the uncertainty of uncharacterized faults. However, the proposed methodology to validate the graph model has not been discussed up to this point. Consider including the evaluation of the model with the proposed metric first. This analysis should reflect the desirable range of the metric and its limitations.

We justify the use of the graph model as a proxy by analysing the ranking correlation between the graph distances and the cumulated mass (Figure 3d). We will precise this around line 145 in the revised manuscript as well as clarify how the proposed metric contribute to assess the performance of the proxy.

- **Results**

[265] In this section, the author should provide a thorough justification of why a metric of 0.3 is considered valid. Based on the plots presented in Figure 5, for a validation coefficient of 0.31, the cumulative mass and the shortest distances seem to differ.

It depends on what we consider a "good" approximation. In line 265, we refer to an "acceptable" threshold. In the case of Figure 5.f, we observe that our method captures two out of the three cumulative mass patches present in the MODFLOW simulation. Indeed, if the user is more demanding, they can choose a higher threshold, such as 0.4 or 0.5. These explanations will be added at the revision of the manuscript.

[272] How does the discretization of the domain affect the binary maps and, consequently, its validation?

Here, we just tested the computing time scalability. We did not compare binary maps of different resolutions. This is something that we can add to the discussion for further work to potentially increase the computing efficiency of the approach.

Figure 5. This figure needs to improve its quality. Consider including the name of the scenario presented in each plot.

The scenario numbers will be added at the revision of the manuscript.

We are providing them now.

Fig a: scenario 0, Fig b: scenario 76, Fig c: scenario 36,

Fig d: scenario 8, Fig e: scenario 27, Fig f: scenario 10

[276] There is no reference to what position 5 is.

Although present in the code, the coordinates of the different positions are indeed not included in the paper. This will be added in the revised version of the paper. We are providing these coordinates for 10 positions, indexed from 0 to 9 :

| ID | X | Y | Z |
| --- | --- | --- | --- |
| 0 | 2011.8216247 | 2950.46369633 | 512.5 |
| 1 | 1644.15961272 | 2948.64944714 | 512.5 |
| 2 | 1811.83145201 | 2423.32644897 | 512.5 |
| 3 | 2327.70259382 | 2409.19913637 | 512.5 |
| 4 | 2049.59368767 | 2027.55911324 | 512.5 |
| 5 | 2253.51310867 | 2538.14331322 | 512.5 |

| | | | |
|---|---|---|---|
| 6 | 1829.7317165 | 2788.42870343 | 512.5 |
| 7 | 1803.19482929 | 2453.49788948 | 512.5 |
| 8 | 1634.04169725 | 2403.11298645 | 512.5 |
| 9 | 1703.45524068 | 2262.31334044 | 512.5 |

We will add such a table at the revision of the manuscript.

Figure 7. This plot references 8 different scenarios from the graph method against one single scenario solved using a physics-based model. In the following paragraph, the author should provide an explanation of why two different scenarios lead to similar or equal validation metrics. This is misleading as it could mean that the proposed validation metric is not robust.

When fault 1 act as a preferential path and fault 2 as a barrier, most of the flow goes through fault 1, which reaches the model outlet independently of fault 3 (that could act either as a barrier or a preferential path). It means that fault 3 does not influence the shortest path through the graph. These explanations will be added at the revision of the manuscript.

Table 2. The caption and names of the scenarios don't match.

Thanks for pointing this out. It is about scenarios 12 and 65 and the caption will be modified accordingly at the revision of the manuscript.

**Technical corrections**

The figures in the manuscript could be significantly improved in terms of clarity and readability. To enhance the visual appeal and understanding of the results. The font size for labels, axis titles, and legends should be increased to improve visibility. Clear and concise labels should be used to identify different components of the figures. Avoid using abbreviations or overly technical terms. Employ distinct color bars for different variables to facilitate comparison and interpretation. Consider the overall layout of the figures, ensuring that the elements are well-organized and easy to follow.

We will update the figures to improve their readability as suggested and such that the font size are increased and are consistent through the different figures of the manuscript.

---

## Author Response (AR1)

**Positive Aspects**

- The graph-based approach simplifies complex geological models and reduces the computational costs.
- Distance map provides information about the potential pathways of contaminant transport.
- A new similarity measure used to compare the distance map to the cumulative mass distribution.

Thanks for your positive assessment of this work.

**General Comments**

• The term "groundwater" is often associated with specific subsurface conditions and flow regimes. While the principles of flow and transport in porous media can be applied to groundwater systems, the broader context of the study seems to be more general. It's important to use more accurate and inclusive terminology to avoid potential misunderstandings, a suggestion could be to use *porous media*.

Our work is motivated by groundwater applications, as stated in the introduction. It is also illustrated with a groundwater synthetic case. While such work could have more general applications to flow and transport in porous media, it has not been tested. Nonetheless it could be tested and applied in other setting.

The following sentence has been added in the last paragraph of the discussion:

"While our work could have more general applications to flow and transport in porous media, it has not been tested yet and could be investigated in future research."

Including fault scenarios might seem unnecessary if the method doesn't perform well for cases without faults, as Appendix A shows.

- Justify the Fault Scenarios: If the fault scenarios are crucial for real-world applications, provide stronger justification. Perhaps there are specific geological settings where faults significantly impact flow and transport.
- Under this specific scenario, explore the limitations of the graph-based approach to justify the range of the metric that is considered acceptable.
- Appendix A needs to include details of parametrization for the MODFLOW simulation.

Exploring conceptual uncertainty such as different fault scenarios is a key motivation of our work, as highlighted in the introduction. As explained in Appendix A, the absence of very high conductivity paths (or very low conductivity barriers), which the graph approximates quite well can explain the mitigated performance of a graph-based approach in a multi-Gaussian setting. So, the use of the method is particularly interesting to tests scenarios displaying different types of hydraulic conductivity contrasts or pathways. In

Appendix A, we already state that the same parameters are used but we could precise 'same flow and transport boundary conditions'.

The following has been added to the second paragraph of the discussion:

"However, the absence of very high conductivity paths (or very low conductivity barriers), which the graph approximates quite well can explain the mitigated performance of a graph-based approach in a multi-Gaussian setting. So, the use of the proposed approach is particularly interesting to tests scenarios displaying strong hydraulic conductivity contrasts or very different pathways."

• The method still relies on a 3D simulation (MODFLOW) to generate the "ground truth" against which the graph-based method is compared. This limits the method's independence and its potential for significant computational savings. While the graph-based method can provide a quick and potentially accurate approximation, perhaps consider validation with simplified Analytical Solutions, Sensitivity Analysis or Machine Learning techniques. This would provide a more rigorous comparison without relying on numerical simulations.

The generation of a synthetic "ground truth" (here using MODFLOW) is indeed necessary to test our approach. In a real case study, is seems reasonable to have monitoring wells at the outlet of the model boundaries, such as to interpolate cumulated mass at the outlet. It would not require the use of flow and transport simulations.

• Similarity measure: A similarity coefficient of 0.3 might seem low, especially considering that a perfect match would be 1.0. While a higher similarity coefficient would be ideal, a value of 0.3 can still be considered reasonable but needs to be explicitly acknowledged, especially given the complexity of the problem. The authors should provide a detailed discussion of the factors influencing the similarity coefficient and explain why this value is acceptable in the context of their study. Additionally, the authors could explore ways to improve the accuracy of the graph-based method, such as refining the graph construction by experimenting with different graph configurations to capture the underlying geological features better.

Lines 263 to 266, we explain how we set this threshold of 0.3. Indeed, the proposed similarity metric is very sensitive to slight changes: a small shift both decreases the Wasserstein component of the similarity and decreases the Jaccard index. The advantage and drawback of each component are given in section 2.4 and justify the proposed metric.

The following paragraph has been added in the discussion:

"The proposed similarity metric tries to mitigate the drawbacks of each of its components. On one hand, the Jaccard index penalizes the comparison of small areas, as a single-pixel error might significantly impact the IoU ratio in that case, and cannot discriminate between non-overlapping scenarios. On the other hand, the NWD penalizes cases where a dissimilar pixel is very far from the areas of similarity between two images. However, one can note that it is very sensitive to slight changes: a small shift both decreases the Wasserstein component of the similarity and decreases the Jaccard index."

- A comprehensive evaluation of the graph-based method requires a clear understanding of the underlying physics-based model, including its setup and initial conditions. The authors should provide a detailed description of the MODFLOW simulations, including:
  - o Model Domain: The spatial extent and discretization of the model domain.

- Hydrogeological Properties: The values assigned to hydraulic conductivity, porosity, and other relevant parameters.
- Boundary Conditions: The types of boundary conditions applied to the model boundaries.
- o Initial Conditions: The initial distribution of hydraulic head and contaminant concentration.

All these characteristics are already described in section 2.1. Figure 2 illustrates sections of hydraulic conductivity for one of the fault scenarios. All scenarios are provided along with the code to generate them as explained in the Code and data availability section (line 430).

Comparing a single MODFLOW scenario to multiple graph-based scenarios can be
misleading, as it doesn't directly assess the accuracy of each individual graph-based
scenario. A more appropriate approach would be to compare each corresponding pair of
scenarios

As stated in lines 260-261, "The similarity index described in Sect. 2.4 has been applied to analyse the results of the 80 scenarios. A representative sample of the results can be found in Fig. 5, and the distribution of similarities is shown in Fig. 6." It means that for each of the 80 scenarios, we compute a similarity index (illustrated in Fig 5) between the MODFLOW cumulative mass map and the GRAPHFLOW distance map. The histogram of the 80 resulting similarity indices is displayed in Fig. 6.

The first paragraph of section 3.1 now reads:

"The similarity index described in Sect. 2.4 has been applied to analyse the results of the 80 scenarios. For each scenario, a MODFLOW simulation is run to obtain the cumulative mass, a graph calculation is performed to obtain a distance map, and the two outputs are compared via the similarity index. A representative sample of the results can be found in Fig. 5, and the distribution of similarities is shown in Fig. 6."

Regarding scenario selection, as explained in section 2.5.1, for a given contaminant source position all pairs of scenarios are tested against each other in the scenario selection. The success rate and number of scenarios are averaged across the 80 possible scenarios. We should precise that the average is over all pairs of scenarios and all contaminant sources.

The beginning of section 3.1 has been reformulated as:

"The results of the average Success rate  $\bar{Y}$ , computed over pairs (80) of fault scenarios (8) and contaminant sources (10), as a function of the average number of selected scenarios..."

- The paper should be understandable to a broad audience without requiring extensive external references. Consider providing a brief explanation of the algorithms used:
  - o Dijkstra's Algorithm
  - Other Algorithms (Jaccard dissimilarity, Wasserstein distance, Otsu thresholding)

We already provide explanations of the Jaccard dissimilarity and Wasserstein distance in section 2.4 as we combine them to propose a new metric. For Dijkstra's algorithm and Otsu Thresholding, simple sentences (around lines 117 and 154 respectively) already describe the purpose of these methods, which is sufficient for the reader.

**Specific Comments**

**• Abstract:**

[2] The phrase "large-scale structural features" could be more specific. Explicitly mention geological features: "large-scale geological features, such as faults, fractures, and stratigraphic variations" and their standard scales compared to domain extension.

The second sentence of the abstract has been updated as suggested.

**• Introduction:**

[42-43] The paper should clearly state how the methodology "improves the consistency for subsurface flow". The author should provide a more precise explanation of why faults are relevant for contaminant transport in porous media. The manuscript should provide a deeper analysis of the role of heterogeneity within the graph-based approach.

The previous work from Rizzo and de Barros (2017) is limited to 2D multi-Gaussian heterogeneous medium and compares the graph approximation with results from particle tracking. Here we go one step further by integrating general flow direction information and by doing a comparison with flow and transport simulations (thus aiming at improving the consistency with subsurface flow).

These precisions have been added in the introduction as:

"To do so, we adapt the approach of Rizzo and de Barros (2017), that is limited to 2D multi-Gaussian heterogeneous medium. Here we go one step further by integrating general flow direction information and by doing a comparison with flow and transport simulations, thus improving its consistency with subsurface flow..."

[47] Consider addressing the role of heterogeneity in the main body of the manuscript.

I am not sure what this comment means. We agree that we should talk more about the role of heterogeneity in the first paragraph of the introduction.

The first paragraph of the introduction as been completed with the following:

"However, these methods often require high computational resources (Karmakar et al., 2022, which restrain the exploration of heterogeneity or geological structural uncertainty, such as faults acting as a preferential flow-path or a barrier, despite their control on flow and transport conditions."

**Method:**

[60] Figure 1. There are no dimensions indicated in the figure. Is there a reason for the orientation of the scheme?

The scheme Has been re-oriented and a scale has been added in the revised manuscript.

[70-73] The description of the experimental setting should be more specific about the position of the source points relative to the grid size. The authors indicate only one coordinate point; it is unclear where the random 10 positions fall on the modeling grid.

The 10 positions are displayed on Figure 2.

A table describing the source point coordinates has been added to section 2.1.

[75-80] This section should also address how the authors evaluate the role of heterogeneity for the simulation domain for the different subsurface properties, as this section indicates a variability in the behavior of the faults but does not answer the effect of the hydraulic conductivity or porosity for this approach. Appendix A should be referenced here.

For each scenario, each geological unit is a stochastic multi-Gaussian SRF realization whose parameters are described lines 78 to 81.

The following sentence has been updated in the last paragraph of section 2.1:

"For each scenario, the hydraulic log-conductivity (before the effects of faults) of each geological unit is modeled by a spatial random field (SRF) with a multi-Gaussian (MG) model..."

[98] Figure 2 shows the hydraulic conductivity values of one scenario. The color bar should be properly labeled, and the formatting of the relative position of the two plots needs to be adjusted.

The label of the colorbar has been added in the revised manuscript.

[100] Equation 2. This equation needs to be properly referenced and described in the text. The variables are not defined.

 $R_{\Gamma}$  is the hydraulic resistance along the path gamma. For each point l on gamma (where l is a dummy variable), we calculate the absolute value of the scalar product between the inverse of the conductivity tensor at point l,  $K^{-l}(l)$ , and the infinitesimal distance dl. We then compute the integral of this value over the path gamma.

This has been clarified in the revised manuscript around equation 2.

[105] Equation 3. This equation needs to be properly referenced and described in the text.

It is the scalar product between the inverse of the hydraulic conductivity simplified tensor  $[k_{xx}, k_{yy}, k_{zz}]$  and the oriented edge  $[e_x, e_y, e_z]$ .

This has been clarified in the text, right after equation 3.

[126] is the function "get shortest paths" the same as the Dijkstra algorithm?

Yes, it is the implementation of Dijkstra algorithm used here.

The first sentence of the second paragraph of section 2.2.2 has been updated to:

"Starting from the weighted and directed graph generated in the previous section, we aim to apply a shortest path algorithm (Dijkstra's algorithm) between the source and the graph nodes corresponding to the model outlet face (for which the hydraulic head is set to 0m on Fig. 1)."

[140] Figure 3. At this stage of the reading, it is still not clear what s32 is. The figure needs quality improvement. Include units for the color bars. Figures c and d should be moved further down as it is not clear at this point what they mean, and they are not formatted properly. Labels for figures c and d should indicate the modeling framework used (MODFLOW, GRAPHFLOW). Furthermore, the

choice of histogram plot to compare the output of 80 simulations using the new methodology compared to one single scenario using MODFLOW is confusing as it does not indicate the performance of each simulation against its corresponding physics-based.

S32 is scenario 32 (numbering is explained lines 73 & 74). Cumulative mass, computed from MODFLOW outputs, is in units of mass per cubic meter (as define line 78) and distances, computed from GRAPHFLOW outputs, are homogeneous to a length in meters, but are not meters as the length through the graph is the product of weighted lengths, so we do not prefer to add a unit to the colorbar of this subfigure, as it could be misleading for the reader.

**Metrics**

[148] Figure 4 needs to improve its quality. Some recommendations: use the same font size of the plots and add labels to the color bars and units of measure. Adjust formatting. Since this is a workflow of the proposed metric, use more descriptive texts next to the figures.

Figure 4 has been reformatted in the revised manuscript.

[178] Variables have different formatting than the previous equation. 2-Wassertein Distance (W2) needs to be numbered.

We have checked the formatting and all 'b's should be bold and italic. The equations formatting, as well as the equations numbering, have been updated in the revised manuscript.

**• Method of scenario selection**

[205-214] This section seems to address a different problem: the uncertainty of uncharacterized faults. However, the proposed methodology to validate the graph model has not been discussed up to this point. Consider including the evaluation of the model with the proposed metric first. This analysis should reflect the desirable range of the metric and its limitations.

We justify the use of the graph model as a proxy by analysing the ranking correlation between the graph distances and the cumulated mass (Figure 3d).

The following sentences have been added and reformulated in the first paragraph of section 2.4:

"The preservation of rank correlation enables to compare areas displaying high values of cumulative mass with areas displaying shortest distances, and suggest that the proposed proxy is relevant. We want to find a metric that spatially compare the pixels in  $I_m$  to the pixels in  $I_d$  with low Dijkstra distances. Ideally, given a number n of pixels in  $I_m$  displaying the highest values of cumulative mass, for a perfect proxy, the pixels in  $I_d$  displaying the n shortest distances would share the same locations in the images."

**• Results**

[265] In this section, the author should provide a thorough justification of why a metric of 0.3 is considered valid. Based on the plots presented in Figure 5, for a validation coefficient of 0.31, the cumulative mass and the shortest distances seem to differ.

It depends on what we consider a "good" approximation. In line 265, we refer to an "acceptable" threshold. In the case of Figure 5.f, we observe that our method captures two out of the three cumulative mass patches present in the MODFLOW simulation. Indeed, if the user is more demanding, they can choose a higher threshold, such as 0.4 or 0.5.

The following text has been added in the second paragraph of section 3.1:

"Note that what can be considered as a valid threshold for a good approximation is subject to the user appreciation. If the user is more demanding, they can choose a higher threshold, such as 0.4 or 0.5. In the case of Fig. 5f, we observe that our method captures two out of the three cumulative mass patches present in the MODFLOW simulation and produces a similarity index of 0.31."

[272] How does the discretization of the domain affect the binary maps and, consequently, its validation?

Here, we just tested the computing time scalability. We did not compare binary maps of different resolutions.

The former 6th, now 7th paragraph of the discussion has been modified with:

"Additionally, it would be interesting to test the scalability of the approach (e.g. by increasing the regular grid resolution or simplifying the graph representation) or other graph algorithms to approximate groundwater flow, to potentially increase the computing efficiency of the approach."

Figure 5. This figure needs to improve its quality. Consider including the name of the scenario presented in each plot.

The scenario numbers have been added in the figure caption.

We are providing them now.

Fig a: scenario 0, Fig b: scenario 76, Fig c: scenario 36,

Fig d: scenario 8, Fig e: scenario 27, Fig f: scenario 10

[276] There is no reference to what position 5 is.

Although present in the code, the coordinates of the different positions are indeed not included in the paper. This will be added in the revised version of the paper. We are providing these coordinates for 10 positions, indexed from 0 to 9:

| ID | X             | Y             | Z     |
|----|---------------|---------------|-------|
| 0  | 2011.8216247  | 2950.46369633 | 512.5 |
| 1  | 1644.15961272 | 2948.64944714 | 512.5 |

| 2 | 1811.83145201 | 2423.32644897 | 512.5 |
|---|---------------|---------------|-------|
| 3 | 2327.70259382 | 2409.19913637 | 512.5 |
| 4 | 2049.59368767 | 2027.55911324 | 512.5 |
| 5 | 2253.51310867 | 2538.14331322 | 512.5 |
| 6 | 1829.7317165  | 2788.42870343 | 512.5 |
| 7 | 1803.19482929 | 2453.49788948 | 512.5 |
| 8 | 1634.04169725 | 2403.11298645 | 512.5 |
| 9 | 1703.45524068 | 2262.31334044 | 512.5 |

The table describing the source point coordinates has been added to section 2.1.

Figure 7. This plot references 8 different scenarios from the graph method against one single scenario solved using a physics-based model. In the following paragraph, the author should provide an explanation of why two different scenarios lead to similar or equal validation metrics. This is misleading as it could mean that the proposed validation metric is not robust.

The following explanations have been added in the first paragraph of section 3.2.

"When fault 1 act as a preferential path and fault 2 as a barrier, most of the flow goes through fault 1, which reaches the model outlet independently of fault 3 (that could act either as a barrier or a preferential path). It means that fault 3 does not influence the shortest path through the graph."

Table 2. The caption and names of the scenarios don't match.

Thanks for pointing this out. It is about scenarios 12 and 65 and the caption has been modified accordingly.

**Technical corrections**

The figures in the manuscript could be significantly improved in terms of clarity and readability. To enhance the visual appeal and understanding of the results. The font size for labels, axis titles, and legends should be increased to improve visibility. Clear and concise labels should be used to identify different components of the figures. Avoid using abbreviations or overly technical terms. Employ distinct color bars for different variables to facilitate comparison and interpretation. Consider the overall layout of the figures, ensuring that the elements are well-organized and easy to follow.

Figures 1, 2 4 and 7 have been updated to improve their readability as suggested and such that the font size are increased.

**Reviewer comments (RC2) in black and answers in blue**

**General Comments**

A new method proposed here would allow a faster ranking of geological multiple scenarios in ground water contamination problems by replacing the grid model per graph and the transport solver within partial differential equations by metrics on graph.

- The details given in the article would allow to reproduce the method by others.
- The article is clearly structured, containing the state of art (introduction), method description and results discussion. The application to a synthetic case is appearing from the beginning of the "Method" part, but since the illustrations on this single application case are helping to understand and to follow the method, it stands well where it is.
- Overall, I appreciate the open discussion where the authors are evocating the remaining challenge of the choice of the threshold or the choice of the particular metrics.

Thanks for your positive comments.

**General Suggestions**

• As far as I understood, the grid is replaced by a graph with no loosing information, where each cell center is replaced by a node and the conductivity between neighbor cells as replaced by a directional edge. Can you state more clearly on that fact in your work, precising that the support of information being change (grid to graph) but with identical information and resolution? Do you use all cells of initial model to create a graph or you neglect the flank cells never participating in the flow? Clarify please that there is no upscaling nor graph reduction here and so it is a perfectly bijective transformation. One it is said, would it mean the heart of your approach is not in grid to graph transformation but in the proxy of flow simulator?

Yes, this is correct, as mentioned at the end of section 2.2.1 ("Thus, we obtain a graph with exactly the same resolution as the original simulation space..."). We can precise that we use all cells of the initial model, keep identical information and resolution, and that there is no upscaling nor graph reduction.

The last paragraph of section 2.2.1 is now reformulated as:

"To build the graph, we use all cells of the initial model, keep identical information and resolution, and do not perform upscaling nor graph reduction. Thus, we obtain a graph with exactly the same resolution as the original simulation space (as many nodes in the graph as cells in the grid representation), with edge weights that accurately approximate the cost for the contaminant to traverse that edge."

• For the same clarity purpose, I would separate the replacement of grid by graph step from the step of replacement of the flow-transport simulator by a proxi with graphs metrics computation. In more general application, the Dijkstra or other graph metrics algorithms may easily by applied to a grid support and get the same results (since the transformation from grid to graph is bijective and finally just a question of format of the data).

Each of these two steps already have its own subsection 2.2.1 Graph generation (for the replacement of grid by graph step) and 2.2.2 Computation (for the step of replacement of the flow-transport simulator by a proxy).

• In case if the transformation to the graph is crucial for this work, please argue this and demonstrate that the following algorithms would not work elsewhere.

Dijkstra's algorithm finds shortest paths between nodes in a weighted graphs. This is why it is crucial to format the geological model as a graph.

The paragraph of section 2.2 has been reformulated as:

"In order to take advantage of Dijkstra's algorithm to find shortest paths between graph nodes and use such a formulation as an approximation for subsurface contaminant flow and transport, the underlying aquifer model has to be represented as a graph. Here we explain how the regular-grid discretization of an aquifer model can be converted into a graph."

• I would place the information in Appendix A in the beginning of the methodology description. As I understood, the proposed approach is performing less good in more homogeneous media. It is not a blocking point itself, but you need to demonstrate that for the other same conditions and the same "matrix" media, your approach do perform differently in the case where you have contract heterogeneities (with and without faults).

We prefer to keep this part in appendix as it does not contribute to address conceptual uncertainty exploration and scenario selection which is the main motivation of this work. As explained in Appendix A, the absence of very high conductivity paths (or very low conductivity barriers), which the graph approximates quite well can explain the mitigated performance of a graph-based approach in a multi-Gaussian setting. So, the use of the method is particularly interesting to tests scenarios displaying different types of hydraulic conductivity contrasts or pathways.

The second paragraph of the discussion has been completed with the following:

"However, the absence of very high conductivity paths (or very low conductivity barriers), which the graph approximates quite well can explain the mitigated performance of a graph-based approach in a multi-Gaussian setting. So, the use of the proposed approach is particularly interesting to tests scenarios displaying strong hydraulic conductivity contrasts or very different pathways."

• The calibration of the threshold on the distance map for your methodology should be done using the parallel with the conventional flow-transport results (with MODFLOW). It is understandable that for the brand-new approach such calibration could be needed. But for the eventual industrial use of your approach, would your approach will depend on the conventional result or you may envisage another calibration process?

We do not envisage other calibration methods at the moment. Nonetheless, the calibration can be conducted on a limited number of scenarios for a specific setting. It would still allow for the exploration of additional scenarios. We would be keen to hear about alternatives.

• The fact that you are using an oriented graph does limit you to apply your approach to the highly connected media? This is the reason why your fractures are not connected to each other in your synthetic example? If such is the case, please discuss it in the limits of your approach application. What would be the challenge if we want to use your approach on the non-oriented graph?

The graph is similar to a non-oriented graph, as all edges are 'duplicated' such that for an oriented edge connecting vertex 1 to vertex 2, and oriented edge connecting vertex 2 to vertex 1 exists. We use oriented edged as a way to integrate general flow information such as the main flow direction.

We added the following after the first paragraph of section 2.2.1:

"Though the graph is built as an oriented graph, it is similar to a non-oriented graph, as all edges are 'duplicated' such that for an oriented edge connecting vertex 1 to vertex 2, and oriented edge connecting vertex 2 to vertex 1 exists. We use oriented edged as a way to integrate general flow information such as the main flow direction."

**Details**

**• Formulas and equations**

In most of the paper formulas and equations one or two terms are not defined in the text. It is quite easy to guess who is who, but it is not homogeneous. You can whether pass through all variables and all text in the article or create a table of annotations in the beginning of the Method paragraph.

We have carefully checked that and updated the manuscript accordingly.

**• 2.1 Experimental settings:**

In real study, if the transmissivity of the fault is unknown, one would define an uncertainty range as a continuous random variable. Would your approach work in this case? Or, because of the efficiency, discussed earlier for the homogeneous media, there are some intermediate situations where it would not work and though would not discriminate the multiple generated cases?

The sensitivity of uncertainty range around fault transmissivity could be tested. But it is likely that in some intermediate situations, the ambiguity would remain and the proxy would not enable to discriminate between different cases. However, such a sensitivity analysis can be solely conducted on the proxy (without running a physical solver), by looking at the sensitivity of different fault transmissivity values on the shortest paths or graph distance maps; it could then be used to define range of values for differentiable scenarios.

**• 2.2.1 Graph generation:**

[99] Figure 2. There is a figure of the conventional grid containing a 3D property. This paragraph is focusing on the graph creation. May you illustrate the resulting graph? or at least a zoom on the peace of the graph?

Given that the graph connects each neighbouring grid cell, it would not identify specific features and not simplify the visualization of the model for the reader, thus we chose not to provide such a plot.

[100] Equation 2. Variables R hydraulic and *dl* are not referenced.

*R* is the hydraulic resistance and *dl* is the incremental length along the path.

These precisions have been added above and below equation 2.

[106] Equation3. Variable Re is not referenced. ...

 $R_{\rm e}$  is the hydraulic resistance of edge e.

The variable has been referenced just above equation 3.

---

## Referee Report (RR1)

**GENERAL COMMENTS**

This manuscript presents an extension of the work by Rizzo and de Barros (2017). The developed surrogate model is tested on an example, where the goal is to determine the permeability character of three faults in a domain for which the conductivity field, the position of the sources and the position of the faults is known.

I was not one of the reviewers of the original submission, and, therefore, I will add some comments to the remarks of the colleagues who reviewed the original submission.

The manuscript is well organized and written with a good language.

However, in my opinion the innovative content of the work is quite limited and the test used to validate the surrogate model is excessively simple.

I think that the work suffers from some scientific weaknesses, that are listed in the specific comments below.

Overall, I think that the manuscript cannot be considered for publication in its present form, but it requires a major revision.

**SPECIFIC COMMENTS**

- 1) Lines 2 & 3. I found this sentence rather obscure and, if I understood it properly, I do not fully agree with the concept it conveys. More generally, at the end of the abstract, I had the idea that the work proposes a surrogate model, based on graph theory, rather than a physically-based model, in order to exclude some hydrostratigraphic or hydrogeological scenarios. However, I am not sure that I got the right point.
- 2) Lines 120 to 124. This remark is intriguing, but I do not understand the physical motivation. Moreover, if I understood correctly, equation (4) implies that the local flow direction is always the same as the main direction of flow. In the case of a highly heterogeneous medium, the geometry of low permeability structures could yield a local flow with a direction opposite to the average flow. Therefore, I am afraid that this formula could not be optimal for some conditions.
- 3) Section 2.3 Observation time. I think that this discussion is not well developed. In particular, the surrogate model proposed here gives a steady-state picture of flow, which is then used to estimate transient solute transport, without solving the ADE. So, the problem of the selection of the observation time should be analysed very carefully.

- 4) Section 2.4 Metrics. The proper selection of metrics for this comparison is a well known problem, for which, to my knowledge, there is no definite solution. the authors provide an interesting discussion, but I would appreciate if they can support their remarks with some numerical tests.
- 5) Lines 224 to 229. The authors assume that the conductivity field and the position of the faults (this is not explicitly stated in the text, but I deduced it from the rest of the description here) are known. Then the problem is restricted to the determination of the permeability o the faults. This problem could be solved quite simply with few runs of a numerical, physically-based model. Indeed, the problem is often that scarce data do not allow to map preferential flow paths (permeable faults, fractures, permeable sediments, etc. for different types of aquifers) and their connectivity.
- 6) In several steps of the work, thresholds are introduced. As it is common in such approaches, the selection of thresholds is often a subjective operation. A thorough sensitivity analysis with respect to the applied thresholds, extending what is found in Appendix B, would be necessary.
- 7) Throughout the whole work, MODFLOW is mentioned as the model applied to generate synthetic data. However, to my knowledge, MODFLOW solves the flow equations, whereas MT3D or other models are used in cascade to model transport. Is this the case? I think this should be specified in a better way.

**TECHNICAL COMMENTS**

- 1) Line 2. What is "it"? the subject of the previous sentence is "groundwater contaminant transport problems".
- 2) Lines 14 to 16. Sentence "The study of... heterogeneous environments" is quite tautological. Sentence "Contamination... mitigation strategies" is quite generic and it ends with a couple of citations that do not seem to be the most relevant to support this statement.
- 3) Line 40. "CO2" should be corrected.
- 4) Line 48. Expression "multi-heterogeneous-layer" should be rephrased.
- 5) Figure 1. The three geological units mentioned at line 67 are not represented. It would be useful to provide the length of the domain along the x and y directions, otherwise using the scale length remains quite imprecise. These values are given at line 79; they could be given earlier.
- 6) Line 70. Sentence "Flow... equation" is rather imprecise. ADE (advection-diffusion equation) is used to model transport and it is based on Darcy's law and on Fick's law for diffusion and dispersion. Moreover, Darcy's law is used to model flow, together

- with the continuity equation (i.e., the mathematical formulation of the mass conservation principle).
- 7) Line 71. The authors mentioned MODFLOW, which is a finite difference code, but here they underline the use of finite elements. OK, this is not a big problem for the specific work presented here, but I'd prefer to avoid confusion. Moreover, these sentences are not well related with the following sentences. And at lines 79 & 80, cell size is mentioned: this seems to be related to a finite difference approximation, rather than to a finite-element simulation.
- 8) Line 72. Increase or decrease K with respect to what?
- 9) Lines 76 & 77, 80 & 81. Measurement units are missing for the coordinates of the reference point (lines 76 & 77) and of the border planes (lines 80 & 81).
- 10) Table 1. The position of wells is given with a precision of 10-8 m, i.e., 10 nm. This is not physically significant!
- 11) Line 81. The average hydraulic gradient is 1/70, which is slightly greater than the typical value of hydraulic gradients, whose magnitude is of the order of 1/1000.
- 12) Line 85. I do not like the "e" format used here to denote values: 3.5×10-5 ms-1 is much better, in my opinion. Measurement units should be attached to each value of a list, as required by rules of the SI system.
- 13) Lines 104 & 105. I do not understand the link between the two sentences "The conductivity field... discrete fields" and "here... 3D space".
- 14) Lines 196 & 205. Using *i* for both the point index and the coordinates might be confusing.

---

## Author Response (AR2)

In blue are our answers to the reviewer reports based on the revised manuscript (first revision).

**Report #1**

Submitted on 09 May 2025

Anonymous referee #1

accepted subject to technical corrections

Overall, I find the response to my comments acceptable, and I value the provided clarification. I have some suggestions that might enhance the manuscript's readability:

Thanks for taking the time to assess the revised manuscript and for your positive and constructive comments.

Table 1. Rounding the numbers to 2 decimals looks better in this case.

We agree that rounding the values improves the readability of the table. We have therefore updated Table 1 so that all values are now rounded to two decimal places.

Line 100: Could you elaborate on the necessity of having duplicated edges in this situation?

As mentioned in line 102, we chose to work with a directed graph in order to capture the predominant direction of the flow. Without this directionality, paths in the graph could propagate information backward, which would not be physically meaningful in our context. Duplicated edges (i.e., edges in both directions) allow the model to learn distinct behaviors depending on the flow direction.

Line 133: Is this meant to be Re or we from Eq 4?

Thank you for pointing this out. You are right, this should be we. We have corrected the manuscript accordingly.

Line 157: Consider moving the reference to the figures (Fig. 3 (a) and (b)) between "mass map \_ over the 80 cases"

Thank you for the suggestion. We have followed your advice and moved the reference to Fig. 3 (a) and (b) accordingly in the revised manuscript.

Figures 3a and 3b do not include units of dimension for the quantities being plotted. For clarity, consider including the coordinate x of the plane being presented, as it is the first time you introduce the points at which the approach is measuring the similarity index.

Thank you for your comment. The axes in Figures 3(a) and 3(b) are given in  $y/\Delta y$  and  $z/\Delta z$  which are dimensionless coordinates corresponding to the number of discretization steps in the y and z-directions. As stated in lines 79 and 135, these maps are plotted at the outlet of the domain (the last layer in the x-direction), located at x=7000 m. We have clarified this more explicitly in the manuscript and updated the figure caption accordingly to avoid any ambiguity.

Figure 7: Specify that this is comparing the 'cumulative mass' of scenario 65 with 'distances' from all scenarios in S5

Yes, we agree that clarifying the caption would help the reader better understand the content of Figure 7. We have revised it accordingly.

**Discussion:**

Line 355: Do the authors have any insights into what is considered a strong difference in the conductivity of the medium? As part of acknowledging the limitations of the proposed approach, is there any sensitivity regarding when the model begins to lean more towards the heterogeneous-no-faults case?

Here, the strong hydraulic conductivity contrasts of two orders of magnitude (or factor 100) is controlled by the definition of the faults. We did not perform a sensitivity analysis of the dividing/multiplying factor on the performance of the proxy.

**Figures:**

Figure captions need to be more descriptive to align with the author's interpretation. Some figures have detailed captions, while others lack sufficient information. Although there's a description in the main text, the figures should be self-explanatory.

Thank you for this valuable comment. We have revised and expanded the captions of several figures to make them more descriptive and self-contained, better reflecting the interpretation provided in the main text. In particular, we updated the captions of Figures 1, 2, and 4 to ensure that they are informative and understandable independently from the main manuscript.

**Report #2**

Submitted on 13 May 2025

Anonymous referee #3

reconsidered after major revisions

**GENERAL COMMENTS**

This manuscript presents an extension of the work by Rizzo and de Barros (2017). The developed surrogate model is tested on an example, where the goal is to determine the permeability character of three faults in a domain for which the conductivity field, the position of the sources and the position of the faults is known.

I was not one of the reviewers of the original submission, and, therefore, I will add some comments to the remarks of the colleagues who reviewed the original submission.

The manuscript is well organized and written with a good language.

However, in my opinion the innovative content of the work is quite limited and the test used to validate the surrogate model is excessively simple.

I think that the work suffers from some scientific weaknesses, that are listed in the specific comments below.

Overall, I think that the manuscript cannot be considered for publication in its present form, but it requires a major revision.

Thanks for providing a review on this revised manuscript. All comments have been carefully addressed point by point.

**SPECIFIC COMMENTS**

1) Lines 2 & 3. I found this sentence rather obscure and, if I understood it properly, I do not fully agree with the concept it conveys. More generally, at the end of the abstract, I had the idea that the work proposes a surrogate model, based on graph theory, rather than a physically-based model, in order to exclude some hydrostratigraphic or hydrogeological scenarios. However, I am not sure that I got the right point.

The beginning of the sentence line 2 has been reformulated to clarify its meaning. You are correct in your understanding of our objectives.

2) Lines 120 to 124. This remark is intriguing, but I do not understand the physical motivation. Moreover, if I understood correctly, equation (4) implies that the local flow direction is always the same as the main direction of flow. In the case of a highly heterogeneous medium, the geometry of low permeability structures could yield a local flow with a direction opposite to the average flow. Therefore, I am afraid that this formula could not be optimal for some conditions.

As explained on line 100, the graph is similar to a non-oriented graph. Depending on weights, a shortest path might have directional components opposite to the main flow direction, which is unrealistic given the steady state boundary conditions. Equation (4) implies that the local flow direction (in the graph) can drift with an angle bounded by the ]-90,90[ interval. An angle of 0 would mean a local flow direction aligned perfectly with the main flow direction.

3) Section 2.3 Observation time. I think that this discussion is not well developed. In particular, the surrogate model proposed here gives a steady-state picture of flow, which is then used to estimate transient solute transport, without solving the ADE. So, the problem of the selection of the observation time should be analysed very carefully.

This is what we acknowledge in section 2.3. We already explain and justify our choices. A sensitivity of the retained observation time is not the scope of this work.

4) Section 2.4 Metrics. The proper selection of metrics for this comparison is a well known problem, for which, to my knowledge, there is no definite solution. the authors provide an interesting discussion, but I would appreciate if they can support their remarks with some numerical tests.

The use of pixels ranked by distances / cumulative mass is supported by the quantitative correlation analysis (Fig. 3). The arguments advanced to build the proposed metric are supported by the literature cited in the section, so we do not deem necessary to add a numerical example.

5) Lines 224 to 229. The authors assume that the conductivity field and the position of the faults (this is not explicitly stated in the text, but I deduced it from the rest of the description here) are known. Then the problem is restricted to the determination of the permeability o the faults. This problem could be solved quite simply with few runs of a numerical, physically-based model. Indeed, the problem is often that scarce data do not allow to map preferential flow paths (permeable faults, fractures, permeable sediments, etc. for different types of aquifers) and their connectivity.

Indeed, we try to identify if faults act as barrier or pathway. The challenge with physically based models is their high computing requirements (to solve the transport - ADE). This is why we are interested to see how good a (faster) proxy can do, as it provides the opportunity to test easily more scenarios.

6) In several steps of the work, thresholds are introduced. As it is common in such approaches, the selection of thresholds is often a subjective operation. A thorough sensitivity analysis with respect to the applied thresholds, extending what is found in Appendix B, would be necessary.

Section 3.3 already provides a sensitivity analysis of the success rate (for scenario selection) as a function of the threshold lambda (applied to the similarity metric).

7) Throughout the whole work, MODFLOW is mentioned as the model applied to generate synthetic data. However, to my knowledge, MODFLOW solves the flow equations, whereas MT3D or other models are used in cascade to model transport. Is this the case? I think this should be specified in a better way.

On their webpage, <a href="https://www.usgs.gov/software/modflow-6-usgs-modular-hydrologic-model">https://www.usgs.gov/software/modflow-6-usgs-modular-hydrologic-model</a> the USGS presents MODFLOW 6 as a container for Groundwater Flow (GWF) and Groundwater Transport (GWT) models. We have update the term MODFLOW to MODFLOW 6 throughout the manuscript.

**TECHNICAL COMMENTS**

1) Line 2. What is "it"? the subject of the previous sentence is "groundwater contaminant transport problems".

The pronoun "it" in line 2 refers to the computational cost associated with groundwater contaminant transport problems, which limits the exploration of conceptual uncertainty. To clarify this, we have rephrased the sentence in the revised manuscript.

2) Lines 14 to 16. Sentence "The study of... heterogeneous environments" is quite tautological. Sentence "Contamination... mitigation strategies" is quite generic and it ends with a couple of citations that do not seem to be the most relevant to support this statement.

We acknowledge that the original sentences were somewhat tautological and generic. We have revised this passage to improve clarity and precision, and we have removed the first reference while keeping the second one. In addition, we have added a relevant reference (Bear and Cheng, 2010) that specifically addresses the modeling of contaminant transport in groundwater systems to better support this context.

3) Line 40. "CO2" should be corrected.

We have corrected "CO2" as requested.

4) Line 48. Expression "multi-heterogeneous-layer" should be rephrased.

We are keeping the original formulation. It is a bit tedious, but it states what it represents: a faulted medium composed of multiple layers, each of them being heterogeneous.

5) Figure 1. The three geological units mentioned at line 67 are not represented. It would be useful to provide the length of the domain along the x and y directions, otherwise using the scale length remains quite imprecise. These values are given at line 79; they could be given earlier.

The three geological units are synthetic and their role in the study is entirely defined by the porosity and conductivity values described in lines 78–79. Their purpose is to introduce heterogeneity relevant for contaminant transport modeling, without referring to real-world lithologies. To improve clarity, we have now added the spatial dimensions of the domain in the figure 1 caption, as suggested. However, we chose to present all the analytical data regarding soil properties together in lines 74–83, in order to keep the description consistent and easily accessible in one block.

6) Line 70. Sentence "Flow... equation" is rather imprecise. ADE (advection-diffusion equation) is used to model transport and it is based on Darcy's law and on Fick's law for diffusion and dispersion. Moreover, Darcy's law is used to model flow, together 3 with the continuity equation (i.e., the mathematical formulation of the mass conservation principle).

We agree that the original sentence was imprecise and could lead to confusion between flow and transport modeling. We have revised the sentence to clearly distinguish between the equations used for flow (Darcy's law with the continuity equation) and those used for transport (the advection–diffusion equation based on Darcy's law and Fick's law). Thank you for pointing this out.

7) Line 71. The authors mentioned MODFLOW, which is a finite difference code, but here they underline the use of finite elements. OK, this is not a big problem for the specific work presented here, but I'd prefer to avoid confusion. Moreover, these sentences are not well related with the following sentences. And at lines 79 & 80, cell size is mentioned: this seems to be related to a finite difference approximation, rather than to a finite-element simulation.

Thank you for pointing this out. You are absolutely right — we mistakenly referred to a finite element solver, while in fact, the numerical simulations were conducted using a finite difference approach, consistent with the use of MODFLOW and the reference to cell size later in the text. We have corrected this in the revised manuscript.

8) Line 72. Increase or decrease K with respect to what?

Thank you for this remark. We agree that the sentence was ambiguous. The increase or decrease in conductivity due to the faults is defined with respect to the value that would be assigned by the underlying multi-Gaussian conductivity field in the absence of faults. We have clarified this point in the revised manuscript.

9) Lines 76 & 77, 80 & 81. Measurement units are missing for the coordinates of the reference point (lines 76 & 77) and of the border planes (lines 80 & 81).

This has been corrected in the revised manuscript.

10) Table 1. The position of wells is given with a precision of 10-8 m, i.e., 10 nm. This is not physically significant!

This has been corrected in the revised version of the manuscript.

11) Line 81. The average hydraulic gradient is 1/70, which is slightly greater than the typical value of hydraulic gradients, whose magnitude is of the order of 1/1000.

It is true, and any choice here would be subjective. A slightly greater gradient requires less time steps for transport simulations and thus lowers computing requirements as well.

12) Line 85. I do not like the "e" format used here to denote values: 3.5×10-5 ms-1 is much better, in my opinion. Measurement units should be attached to each value of a list, as required by rules of the SI system.

We thank the reviewer for this suggestion. We agree that the scientific notation with "×10-5" and proper unit formatting is clearer and more consistent with SI conventions. We have corrected this notation in line 85 and throughout the manuscript to follow this format

13) Lines 104 & 105. I do not understand the link between the two sentences "The conductivity field... discrete fields" and "here... 3D space".

We agree that the link between the two sentences could be clearer. While physics-based solvers like MODFLOW can handle discrete conductivity fields defined on non-regular grids (e.g., grids with variable cell sizes), in our work, we focus exclusively on regular grids within a bounded 3D space. We have revised the manuscript accordingly to clarify this point.

14) Lines 196 & 205. Using i for both the point index and the coordinates might be confusing. We thank the reviewer for pointing this out. In the original version, the index i always referred to the point index, not to a coordinate component. The expressions xi represented the 2D coordinates of the i-th pixel. However, to avoid any possible confusion, we have revised this section by renaming the variable x by the variable z= {x,y} and we now explicitly state that for each i, zi={xi,yi} denotes the coordinates of the i-th pixel in the image. This clarification should remove any ambiguity regarding the use of indices and coordinate notation.